# Design and Verification of Multi-Agent Systems with the Use of Bigraphs

**Piotr Cybulski ***  **and Zbigniew Zieliński** 

Faculty of Cybernetics, Military University of Technology, ul. gen. S. Kaliskiego 2, 00-908 Warsaw, Poland; zbigniew.zielinski@wat.edu.pl
* Correspondence: piotr.cybulski@wat.edu.pl

**Featured Application: Rapid development of behavior policies for agents in a controlled environment.**

**Abstract:** Widespread access to low-cost, high computing power allows for increased computerization of everyday life. However, high-performance computers alone cannot meet the demands of systems such as the Internet of Things or multi-agent robotic systems. For this reason, modern design methods are needed to develop new and extend existing projects. Because of high interest in this subject, many methodologies for designing the aforementioned systems have been developed. None of them, however, can be considered the default one to which others are compared to. Any useful methodology must provide some tools, versatility, and capability to verify its results. This paper presents an algorithm for verifying the correctness of multi-agent systems modeled as tracking bigraphical reactive systems and checking whether a behavior policy for the agents meets non-functional requirements. Memory complexity of methods used to construct behavior policies is also discussed, and a few ways to reduce it are proposed. Detailed examples of algorithm usage have been presented involving non-functional requirements regarding time and safety of behavior policy execution.

**Keywords:** multi-agent systems; bigraphs; design; verification; modeling; non-functional requirements



## 1. Introduction

With the increase of computational power and its availability comes the desire to incorporate it more into our daily life. Current ideas on how to do this include the Internet of Things, multi-agent systems (in which particular cases are swarms of robots), or smart objects and places (e.g., cities, homes, cars). All of them require new ways to design large-scale (i.e., consisting of a significant number of elements) software and physical systems that consider both how individual components interact and how a system as a whole works. There are various unresolved problems related to this. There is no consensus on what elements of the real world should be modeled and which of their capabilities should be taken into account in general. What is worse, among different design methods elements of the real world are used differently. Finally, the results of these methods are often incomparable, or at least, there is no common way to evaluate multi-agent system design methods. Regardless, any method for designing complex systems must offer a specific range of capabilities to be considered useful.

The concept of agent is applied to entities that have autonomy and are placed in a changing environment. Multi-agent systems [1,2] are structures within which agents can be identified. One of the advantages of designs using agents is that they can be represented at different levels of detail, from abstract entities (like mathematical structures) to actual robots. For this reason, among others, the concept of multi-agent system is used in various contexts. This term may be used to characterize a group of machine learning methods [3,4].

It can also be used to highlight attributes of certain models and simulation approaches [5–7]. The term also refers to a subgroup of robotics solutions [8–11] that make use of widely understood autonomous robots to perform assigned tasks. In this work, we will focus on multi-agent robotic systems (MARS). The literature [12–16] is replete with examples of various applications of multi-agent robotic systems. There are also methodologies and tools [10,17] to design such systems. There is no consensus on how to design such systems in general and current solutions come from different areas of science. The most common paradigms used to design MARS include software design patterns [16], control theory [12,13], optimization theory or combinations of the above [15]. Some examples are utilizing mathematical logic in MARS design [18], but they are much less common. Due to the lack of agreement on how to design MARS and the fact that results produced by different methodologies are difficult to compare, we will try to evaluate them based on their capabilities. In this paper, we will be interested not so much in how to design MARS but rather how the following questions can be answered about an existing project:

- Is the project correctly designed? We want to assure the syntactic correctness, i.e., the correct use of formal tools such as mathematical logic, differential equations, or pi-calculus. We also care about semantic correctness, i.e., the ability to transform a formal model into a real solution (implementable on robots).
- How does one perform a simulation illustrating MAS operation?
- Have non-functional requirements been met? Those regarding safety and speed of task execution in particular.

Verifying the correctness of a model is the simplest and most solutions can be verified using the tools they were made with. Verifying whether a designed system accomplishes a given task is much more difficult. The vast majority of methodologies in the literature use simulation for this purpose. Exceptions can be found among models that highly formalize the internals of agents, how they operate, and the course of a task itself. Verification by simulation also gets complicated as the model becomes more abstract. The simplest designs in this regard are those based on methods commonly used in other areas of science (such as differential equations or graph theory) or made using tools integrated with a simulator. Verification of non-functional requirements is a difficult part of the design. Methodologies commonly found in the literature such as RE4Gaia [19], TROPOS [20], DIAMOND [21], or Adelfe [22] take into account non-functional requirements during design process. They usually aim to enable design of multi-agent systems in general (not just multi-agent robotic systems). Successive stages in most of these methodologies are not closely coupled together. By loosely coupled process, we understand a design process where a designer's interpretation of how the system works plays a significant role the whole time. In other words, one cannot treat the results of one stage as an input that the next stage will automatically transform into a form acceptable by yet another stage. When it comes to verification of system requirements, it should be noted that none of the above methodologies offer formal guarantees regarding the system's functionality as the methods dedicated to specific tasks do. An example of a such method can be found in [13] where a formal guarantee is given for robots to move keeping at least a specified distance from each other (an example of a non-functional requirement). In [12] a guarantee of fulfillment of functional requirements is presented where a task is guaranteed to be carried out if certain conditions are satisfied.

Using bigraphs [23] to design multi-agent systems is a relatively new approach to modeling this kind of system. The bigraph theory was published by Robin Milner in 2008 but has already been extended with a notion of overlapping locations [24] and probability [25]. Bigraphs are currently found useful in areas such as system of systems design [26], IoT [27], and wireless network modeling [28]. Currently, there are a few tools that support modeling systems with bigraphs, the most notable of them are Bigraphical Model Checker [29] (discontinued), Bigraph Framework for Java [30], and BigraphER [31]. The first two of them focus on checking the reachability of certain states of a system [29,31]. At the same time, the last one provides means to analyze various aspects of a modeled system (especially useful in this regard is

underlying OCaml library *bigraph*). We believe that Bigp+rahER [31] provides the most advanced set of utilities to model systems with bigraphs available at the moment. Multi-agent systems design methodologies [32,33] involving bigraphs are scarce, and most of them do not consider generating behavior policies based on a constructed model. As an exception to this, one may point out BigActor methodology described in [34] that uses bigraphs mixed with the notion of actors [35] or our methodology [36] based on bigraphs with tracking.

In [36] we have proposed a methodology based on bigraphs with tracking [23] that enables design of multi-agent systems. We have chosen tracking bigraphs primarily because they allow for analysis of objects' activities over time without introducing another layer of abstraction (as it was done, for example, in [34]). Our methodology is devoid of some of the drawbacks we mentioned earlier, such as loose coupling between design stages or the designer's interpretation of systems internals on all stages of the design process. Moreover, successive stages of the methodology are module-like which means their implementations can be adjusted to project needs. The methodology's main disadvantages are high computational complexity, limitation of system's agents to entities that can be fully controlled, and the fact that the operation of a designed system is determined before it is started. It also does not offer universal guarantees of task successful completion as presented in [12,13,18]. Putting our work in a broader context, we can place our methodology in a group of bottom-up [37] methods of MAS design with a note that it focuses on global goals rather than individual ones. In fact, agents in our approach do not have preferences that can affect their actions. A distinguishing feature of our proposition is the lack of abstractions outside the bigraphs framework, typically agents' internal mechanics are modeled with BDI (Belief, Desire, and Intention) [32,38] or actors [34].

This work is an extension of the methodology proposed in [36]. This paper aims to demonstrate how to verify the correctness of a design, check the fulfillment of non-functional requirements, and visualize behavior policies. We have developed an algorithm to automatically verify the correctness of a model and construct successive simulation states. We also described how to verify whether non-functional requirements are satisfied by a behavior policy for agents in the system. An example implementation [39] of the algorithm has been prepared. We also addressed the memory complexity of operations performed during behavior policy generation. We discussed how it influences the feasibility of projects and suggested a few ways to reduce the memory complexity. Finally, a tool [40] has been implemented that incorporates all of the mentioned memory complexity reduction strategies and a tool [41] to illustrate constructed behavior policies.

## 2. Methods and Materials

In this section, we will introduce all terms and definitions that are necessary to understand examples presented in Section 3. Section 2.1 is devoted to basic informal definitions that will be used throughout the rest of this article. Sections 2.2–2.4 aim to quickly acquaint the reader with the methodology described in detail in [36] and for that reason micro-examples are included at the end of each of these subsections. Section 2.5 is dedicated to an algorithm for verification and visualization of behavior policies. Since the algorithm is the key of this article, examples of its usage are presented in Section 3.

### 2.1. Basic Concepts

Before formal definitions, we will introduce the following concepts:

- Task—A collection of objects from the real world along with the actions they can perform, the initial state, and the target-desired (final) state(s). An example of a task might be:
  "In an area that is a $3 \times 3$ grid, there are two robots in opposite (diagonally) cells. Each robot can move to vertically and horizontally adjacent cells and connect to a second robot if both are in the same cell. The goal of the task is for both robots to connect with each other."
- Mission—a realization of a task.

- Task element—a real-world entity that is relevant to the subject matter being modeled. Elements can be people, robots, areas, data sources, and receivers, etc.
- Passive object—a task element that can participate in activities without initializing them. It may contain other passive objects. We are not interested in their behavior, but we take into account the passage of time for them. The number of passive objects is constant during a mission.
- Active object (agent)—a task element that can participate in activities by initializing them. It can contain other active and passive objects. We are interested in their behavior, and we take into account the passage of time for them. We can control them. It is assumed that the number of agents during a mission is constant.
- Environment—a task element that can participate in activities without initializing them. It can contain passive and active objects and be owned by at most one other object. We are not interested in its behavior, and <u>do not</u> consider the passage of time for it.
- Behavior Policy—A set of planned actions for all agents that meets the following requirements:
  - Implementing a behavioral policy solves a given task;
  - All agents start the mission at the same time;
  - Agents can complete a mission at different points in time;
  - All agent activities must be performed continuously (without time gaps);
  - All agents that participate in a cooperative activity must start performing it at the same moment.
- Scenario—Mission using a specific behavioral policy.

### 2.2. Bigraphs

Through this article we will extensively use bigraphs, *concrete bigraphs* to be precise. Concrete bigraphs allow identifying its nodes and edges with *support* (more about that later). In contrast, *abstract bigraphs* lack the mentioned identifiers. In the rest of this article, whenever we refer to a bigraph, we will have a concrete bigraph in mind. A bigraph consists of two graphs: a place graph and a link graph. Place graph is intended to model spatial relations between system elements. A link graph is a hypergraph that can be used to model interlinking between the elements.

Formally a bigraph is defined as:

$$B = (V_B, E_B, ctrl_B, G_B^P, G_B^L) : I \to O$$

- $V_B$—a set of vertices identifiers;
- $E_B$—a set of hyperedges identifiers. A union of both of these sets makes the *bigraph support*;
- $ctrl_B : V_B \to K$—a function assigning a control type to vertices. $K$ denotes a set of control types and is called a signature of the bigraph;
- $G_B^P = \langle V_B, ctrl_B, prnt_B \rangle : m \to n$ and $G_B^L = \langle V_B, E_B, ctrl_B, link_B \rangle : X \to Y$ denote a place and a link graph respectively. A $prnt_B$ function defines hierarchical relations between vertices, roots, and sites. A $link_B$ function defines linking between vertices and hyperedges in the link graph;
- $I = \langle m, X \rangle$ and $O = \langle n, Y \rangle$ denotes the inner face and outer face of the bigraph $B$. By $m, n$ we will denote sets of preceding ordinals of the form: $m = \{0, \dots, m-1\}$. Sets $X$ and $Y$ represent inner and outer names respectively. When any of the elements of an interface is omitted it means it is either equal to 0 (when interface lacks an ordinal) or it is empty (when there is no set of names). For example, interface $I = m$ means it has no inner names.

An example of graphical representation of a bigraph is presented in Figure 1.

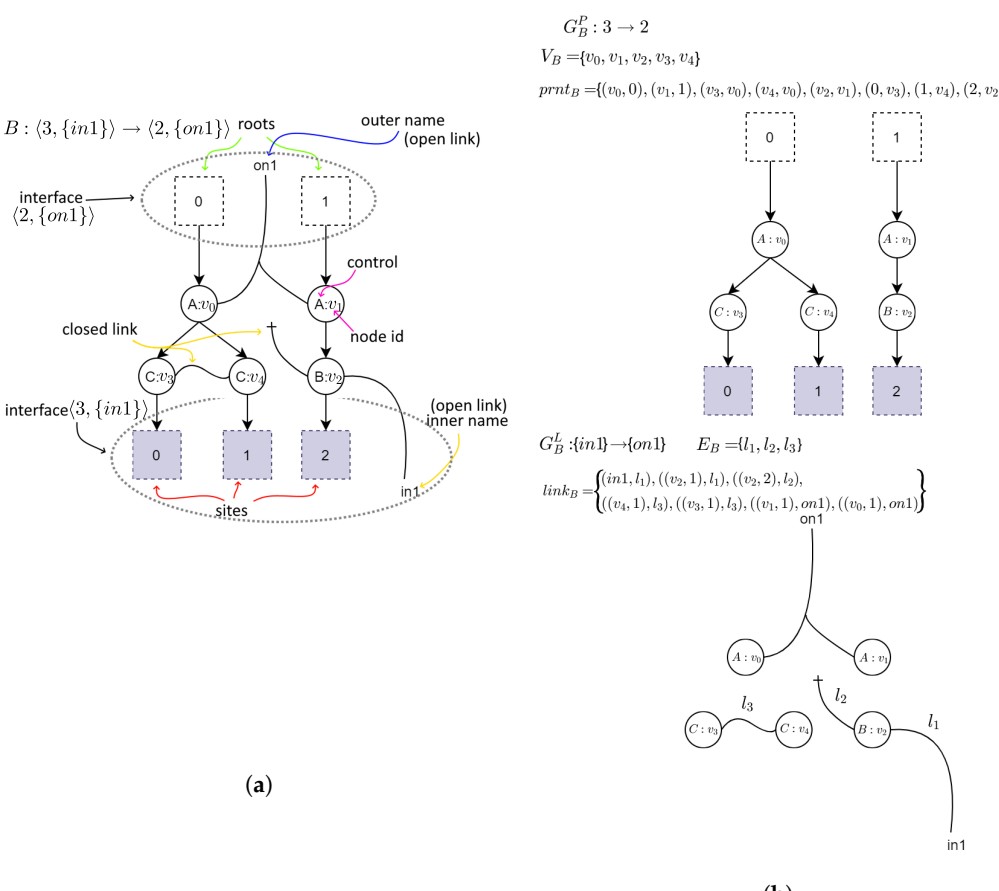

**Figure 1.** An example of a bigraph and its constituents. The right part represents a place graph (the upper part of the figure) and a link graph (the lower part of the figure). They share a signature which defines control types (letters in nodes) and arity of each control (number of unique links that can be connected to a node with specified control). Ports and inner names can be attached to either edges or outer names, that is why there are only three edge identifiers in the link graph. On the left there is the bigraph made from the superposition of them both. (**a**) A bigraph. (**b**) A place graph and a link graph.

Reaction rules are used to model dynamics in bigraphical systems. In this paper, we will use (simplified) tracking reaction rules. Reaction rule consists of a pattern (redex) to be found in an input bigraph that shall be replaced with another bigraph (reactum).

Formally, a tracking reaction rule is a quadruple:

$$(B_{redex} : m \to O, B_{reactum} : m' \to O, \eta, \tau)$$

where:

- $B_{redex}$—a bigraph called redex;
- $B_{reactum}$—a bigraph called reactum;
- $\eta : m' \to m$—a map between sites from reactum to sites in redex;
- $\tau : V_{reactum} \to V_{redex}$—a map of reactum's node identifiers onto redex's node identifiers. It allows one to indicate which elements of an input bigraph are "residues" in an output bigraph.

*Bigraphical Reactive System* (BRS) is a tuple $(\mathcal{B}, \mathcal{R})$ where $\mathcal{B}$ denotes a set of bigraphs with empty inner face and $\mathcal{R}$ is a set of reaction rules defined over $\mathcal{B}$. If $\mathcal{R}$ consists of rules with tracking then a pair $(\mathcal{B}, \mathcal{R})$ makes a *Tracking Bigraphical Reactive System* (TBRS).

Having a TBRS we can generate a Tracking Transition System (TTS). A *Tracking Transition System* is a 7-tuple: $\mathcal{L}_{\mathcal{T}} = (Agt, Red, Lab, Apl, Par, Res, Tra)$ where:

- $Agt$—a set of bigraphs;

- *Red*—a set of redexes used to construct the TTS;
- *Lab*—a set of labels;
- Apl $\subseteq$ Agt $\times$ Lab—an applicability relation;
- *Par* : $V_r^{V_b}$   $r \in Red, b \in Agt$—a participation function. It indicates which vertices in an input bigraph correspond to elements in the redex of a transition;
- *Res* : $V_{b_1}^{V_{b_2}}$   $b_1, b_2 \in Agt$—a residue function. It maps vertices in an output bigraph that are residue of an input bigraph to the vertices in the input bigraph;
- *Tra* $\subseteq Apl \times Agt \times Par \times Res$—a transition relation.

As we said at the beginning of this section, we will use a simple example to illustrate how the formal definitions can be used in practice. The system for our example consists of two areas and two agents (we do not care whether they are humans, robots, or other autonomous entities). Areas will be denoted by controls *A* and *B* while agents will be represented with controls *U*. We assume that agents can move from an area of type *A* to an area of type *B* in two ways, which differ in execution speed. Thus Tracking Bigraphical Reactive System of the system above consists of three bigraphs and two reaction rules. The elements of $\mathcal{B}$ set are described in Table 1 and the reaction rules are defined in Table 2. The Tracking Transition System of this TBRS is defined in Table 3.

**Table 1.** Elements of the $\mathcal{B}$ set for the introductory example.

| Graphical Representation | Name | Description |
|:---:|:---:|:---|
| | s0 | The initial state of the system. |
| | s1 | The state where only one of the agents has moved to the *B* area. |
| | s2 | The state where both agents has moved to the *B* area. |

**Table 2.** Elements of the $\mathcal{R}$ set for the introductory example. The $\eta$ function for the first rule and both $\tau$ functions are identities. The first rule represents an action that allows a single agent to move between areas. The second rule is for an action where two agents move both at once. The second rule is only reasonable if underlying mechanism differs to that of the first rule.

| Graphical Representation | Name |
|:---:|:---:|
| 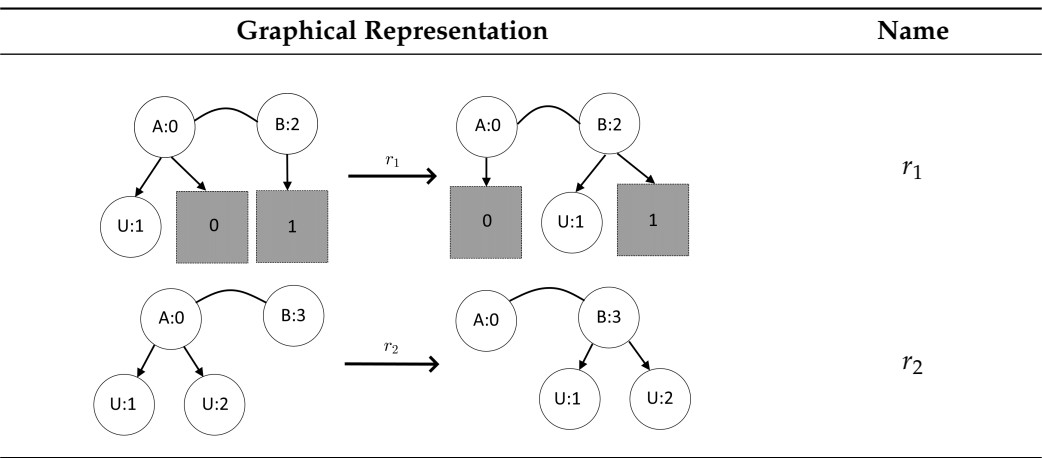 | $r_1$ |
| | $r_2$ |

**Table 3.** The Tracking Transition System for the introductory example. Each row defines a single transition in the system.

| Apl | Agt | Par | Res |
|---|---|---|---|
| $\langle s0, r_1 \rangle$ | s1 | $\{(0,0),(1,1),(3,2)\}$ | $\{(0,0),(1,2),(2,3),(3,1)\}$ |
| $\langle s0, r_1 \rangle$ | s1 | $\{(0,0),(2,1),(3,2)\}$ | $\{(0,0),(1,1),(2,3),(3,2)\}$ |
| $\langle s0, r_2 \rangle$ | s2 | $\{(0,0),(1,1),(2,2),(3,3)\}$ | $\{(0,0),(1,3),(2,1),(3,2)\}$ |
| $\langle s1, r_1 \rangle$ | s2 | $\{(0,0),(1,1),(2,2)\}$ | $\{(0,0),(1,2),(2,3),(3,1)\}$ |

*2.3. State Space*

Having a Tracking Transition System we can transform it into a state space of the modeled system. A state space can be later used to generate a behavior policy for agents (as defined in Section 2.1) in the system.

We assume the following about modeled systems:

1. A number of passive and active objects is constant during whole mission;
2. A system cannot change its state without an explicit action of an agent (alone or in cooperation with other agents);
3. No actions performed by agents are subject to uncertainty;
4. A mission can end for each agent separately in different moments. In other words, agents do not have to finish their part of the mission all at the same time;
5. In case of actions involving multiple objects (whether these are active or passive), it is required of all participants to start cooperation at the same moment.

A state space *SS* of a system consisting of $n_o$ objects and $n_s$ states is defined as:

$$\mathcal{SS} = (S, E, L, I, C, T, M_f)$$

where:

- $S \subset \mathbb{N}$—a set of states in the state space. It corresponds to bigraphs in the Tracking Transition System;
- $E \subseteq S \times S$—a multiset of ordered pairs of states. Elements in this set are directed edges representing transition relations between states;
- $L$—a set of labels of changes in the system. It will usually consist of reaction rule names from the Tracking Transition System the state space originated from. To determine what changes, in what order, have led to a specific state we will additionally introduce set $H = \{l_t | l \in L, t \in \mathbb{N}\}$. Elements of the $H$ set indicate what action (label) took place in what order (index value).
- $I = \{\mathbb{N}_1^2 \times \cdots \times \mathbb{N}_{n_a}^2\}$—a set of possible state-at-time (SAT) configurations. The interpretation of elements in such a set is as follows. The first element in each of inner tuples denotes *id* of an object (either passive or active) in the system. The second element in each inner tuple is meant to represent time at which the object specified by the *id* is at. For example, for $n_o = 2$ the element $i_x = \langle(1,777),(2,123)\rangle$ denotes a situation where the object with id 1 is at the moment 777 while the object with id 2 is at the moment 123.
- $C = (I \times 2^H) \cup \{0\}$—a set of possible mission courses. 0 denotes the neutral element, i.e., $\forall_{x \in C} x + 0 = 0 + x = x$. For the rest of the elements of $C$ set the $+$ symbol serves only as an associative conjunction operator and does not denote any meaningful operation. In other words for the rest of the elements the following formula is true: $\forall_{x,y \in C \setminus \{0\}} x + y = y + x$.
- $T = \{f_i : C \times \mathbb{N} \to C | i \in \mathbb{N}\} \cup \{f_{null}\}$—a set of functions defining progress of a mission. The $f_{null}$ function returns 0 regardless of input. Additionally, we will denote by $T_{i,j} \subset T$ a set of all mission progress functions from the $i$ state to the $j$ state.
- $M_f : E \to T$—a bijective mapping of edges to mission progress functions.

Going back to our introductory example, we will now convert the Tracking Transition System from Table 3 into a state space of the system. We will not define all of the formal elements and rather focus on the key ones. The $\mathcal{S}$ consists of three elements $\mathcal{S} = \{0, 1, 2\}$ that correspond to bigraphs s0,s1 and s2 respectively. The $L$ consists of two elements that correspond to reaction rules in TBRS i.e., $L = \{r1, r2\}$. Knowing that there are only two agents in the system (so there are two objects in total) elements of the set $I$ will be of the form $\langle\langle i_1, x\rangle, \langle i_2, y\rangle\rangle$. The elements $i_1, i_2$ of a tuple correspond to identifiers of objects (in this case $i_1, i_2 \in \{1, 2\}$) and $x$ and $y$ elements indicate a moment of time at which each object is at. We will clarify how to utilize the $C$ set in the next subsection. As it was mentioned earlier, the action represented by the $r_1$ reaction rule takes 2 units of time while the $r_2$ reaction takes only 1 unit of time. How these values are obtained depends on a project and may be subject to many factors such as resolution of time need to be considered (whether these are minutes, seconds or hours) or variability (or lack thereof) of time needed to execute actions represented by reaction rules. Knowing this, the elements of the $T$ set are listed in Table 4. Subsequent elements of this set correspond to transitions in TTS. The permutation being a result of application of a transition function corresponds to permutation of vertices corresponding to objects in *res* function. It is also worth noting that $f_3$ function requires both agents to be at the same time (variable $z$) in order to return something other than 0.

**Table 4.** Mission progress function definitions for the state space presented in Figure 2. The action represented by $r_1$ reaction rule is assumed to take 2 units of time while the action $r_2$ takes only 1 unit of time.

| Function | Function Definition | |
|:---:|:---:|:---:|
| $f_1$ | $f_1(c, t) = \begin{cases} [\langle(b, y), (a, x + 2)\rangle, H' \cup \{r1_{t+1}\}] \\ 0 \end{cases}$ | $\begin{aligned} &: c = [\langle(a, x), (b, y)\rangle, H'] \\ &: c = 0 \end{aligned}$ |
| $f_2$ | $f_2(c, t) = \begin{cases} [\langle(a, x), (b, y + 2)\rangle, H' \cup \{r1_{t+1}\}] \\ 0 \end{cases}$ | $\begin{aligned} &: c = [\langle(a, x), (b, y)\rangle, H'] \\ &: c = 0 \end{aligned}$ |
| $f_3$ | $f_3(c, t) = \begin{cases} [\langle(a, z + 1), (b, z + 1)\rangle, H' \cup \{r2_{t+1}\}] \\ 0 \end{cases}$ | $\begin{aligned} &: c = [\langle(a, z), (b, z)\rangle, H'] \\ &: c \neq [\langle(a, z), (b, z)\rangle, H'] \end{aligned}$ |
| $f_4$ | $f_4(c, t) = \begin{cases} [\langle(b, y), (a, x + 2)\rangle, H' \cup \{r1_{t+1}\}] \\ 0 \end{cases}$ | $\begin{aligned} &: c = [\langle(a, x), (b, y)\rangle, H'] \\ &: c = 0 \end{aligned}$ |

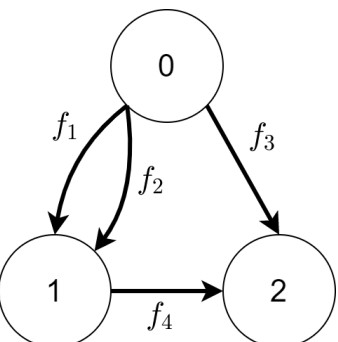

**Figure 2.** The state space generated from Tracking Transition System defined in Table 3. Mission progress functions definitions are defined in Table 4.

## 2.4. Behavior Policy

We define a behavior policy as a schedule of actions for each object from the beginning of a mission to its end that meets all the requirements listed in Section 2.1.

Having a state space, we can view a behavior policy as a walk (in graph theory sense) indicating what changes (and who did them) are required in order to reach a desired state.

To construct a proper policy behavior based on a state space, we need to define the following elements. Please note that by series we will understand a <u>finite</u> sum of elements.

- $K_s^t = c_1 + \cdots + c_m = \sum_{i=1\cdots m} c_i \quad c_i \in C, s \in \{0, \cdots, n_s - 1\}, t \in \mathbb{N}$—a series, where summands are mission courses leading to the state $s$;
- $N_K(K_s^t) \in \mathbb{N}$—a function returning a number of elements in a given series. According to the earlier definition, for any series $K_s^t$ this function returns a value of $m$ (the greatest index of $c_i$);
- $F_{i,j}(x,t) = \sum_{k \in T_{i,j}} f_k(x,t) \quad i,j \in \{0, \cdots, n_s - 1\}, t \in \mathbb{N}$—a series, whose summands are mission progress functions from the $i$ to the $j$ state;
- $\mathbb{M}_K^t = \begin{bmatrix} K_0^t & \cdots & K_{n_s-1}^t \end{bmatrix}, t \in \mathbb{N}$—a matrix whose elements are series indicating possible walks leading to each state. Index $t$ denotes a number of steps made in a state space. By a step we understand a transition between vertices (including the situation where traversal does not change the vertex);
- $\mathbb{M}_F^t = \begin{bmatrix} F_{0,0}(x,t) & \cdots & F_{0,n_s-1}(x,t) \\ \cdots & \cdots & \cdots \\ F_{n_s,0}(x,t) & \cdots & F_{n_s-1,n_s-1}(x,t) \end{bmatrix}$—a matrix of transitions between states.

Furthermore, we define two operations:

- $K_s^t \circ F_{i,j}(x,t) = \sum_{k \in T_{i,j}} \sum_{l=1\cdots N_K(K_s^t)} f_k(c_l, t)$—a convolution of the series defined above;
- $\mathbb{M}_K^{t+1} = \mathbb{M}_K^t \cdot \mathbb{M}_F^t$—a multiplication of the matrices defined above. Elements of the new matrix are defined by the formula:

$$K_s^{t+1} = \sum_{k=0}^{n_s-1} K_k^t \circ F_{k,s}(x,t)$$

In order to generate all walks consisting of a specified number of steps from an initial state to a final state one must define the initial state, as a $\mathbb{M}_K^0$ matrix and multiply subsequent results by $\mathbb{M}_F^i$ the specified number of times. The result will be a $\mathbb{M}_K^x$ matrix, whose summands in the $i$th column will indicate all possible walks with $x$ steps that end in the $ith$ state of the state space. If the element in the specified column is equal to 0, it means there is no such walk.

Summarizing our introductory example, we will demonstrate how to use the state space from Figure 2 with transition functions definitions listed in Table 4 to determine all sequences of actions that lead to the state denoted as s2. Each sequence is equivalent to behavior policy that, when applied, results in moving both agents to the area of type $B$.

To determine such sequences, we create two matrices, a matrix of transitions $\mathbb{M}_F^t$ and matrix of initial state $\mathbb{M}_K^0$. Having both of them, we can multiply subsequent $\mathbb{M}_K^t$ matrices by corresponding $\mathbb{M}_F^t$ matrices and check whether the third state (recall that numbering starts from 0) is reachable. By reachable, we understand having a value other than 0 in the specified column of the $\mathbb{M}_K^t$ matrix.

Definitions of both matrices are listed below:

$$\mathbb{M}_F^t = \begin{bmatrix} f_{null} & f_1 + f_2 & f_3 \\ f_{null} & f_{null} & f_4 \\ f_{null} & f_{null} & f_{null} \end{bmatrix}$$

$$\mathbb{M}_K^0 = \begin{bmatrix} [\langle (1,0), (2,0) \rangle, \varnothing] & 0 & 0 \end{bmatrix}$$

The $\langle (1,0), (2,0) \rangle$ tuple in the first column of $\mathbb{M}_K^0$ matrix denotes that we have two objects. The zeros in both tuples indicate that each object starts the mission at the same moment.

Subsequent $\mathbb{M}_K^t$ matrices allow us to determine how a system changes when a specified number of actions occur. For example, $\mathbb{M}_K^1$ gives us information about how the system evolves when one action occurs (analogously $\mathbb{M}_K^2$ for two actions etc.).

In our example $\mathbb{M}_K^1$ and $\mathbb{M}_K^2$ are of the form:

$$\mathbb{M}_K^1 = \mathbb{M}_K^0 \cdot \mathbb{M}_F^0 = \begin{bmatrix} [\langle (1,0),(2,0)\rangle, \varnothing] & 0 & 0 \end{bmatrix} \cdot \begin{bmatrix} f_{null}(c,0) & f_1(c,0) + f_2(c,0) & f_3(c,0) \\ f_{null}(c,0) & f_{null}(c,0) & f_4(c,0) \\ f_{null}(c,0) & f_{null}(c,0) & f_{null}(c,0) \end{bmatrix}$$

$$\mathbb{M}_K^1 = \begin{bmatrix} 0 & [\langle (2,0),(1,2)\rangle, \{r1_1\}] + [\langle (1,0),(2,2)\rangle, \{r1_1\}] & [\langle (1,1),(2,1)\rangle, \{r2_1\}] \end{bmatrix}$$

$$\mathbb{M}_K^2 = \mathbb{M}_K^1 \cdot \mathbb{M}_F^1 = \mathbb{M}_K^1 \cdot \begin{bmatrix} f_{null}(c,1) & f_1(c,1) + f_2(c,1) & f_3(c,1) \\ f_{null}(c,1) & f_{null}(c,1) & f_4(c,1) \\ f_{null}(c,1) & f_{null}(c,1) & f_{null}(c,1) \end{bmatrix}$$

$$\mathbb{M}_K^2 = \begin{bmatrix} 0 & 0 & [\langle (1,2),(2,2)\rangle, \{r1_1, r1_2\}] + [\langle (2,2),(1,2)\rangle, \{r1_1, r1_2\}] \end{bmatrix}$$

The interpretation of each of the above $\mathbb{M}_K^t$ matrices is as follows. The $\mathbb{M}_K^1$ matrix indicates that with just one action there are two ways for the system to be in the state where one of the agents move to the area of type *B* and the other one will not take any action (as it is pointed out by the fact that its time is equal to 0). Both ways require specified agent to carry out the action represented by the *r1* rule. The same matrix also gives us information that with one action there is a possibility to reach s2 state if both agents engage in cooperative execution of *r2* rule. Finally, the $\mathbb{M}_K^2$ points out two walks in the state space that lead to the s2 state. Both involve performing the action associated with *r1* rule two times (each time by a different agent).

It is worth pointing out that in a software implementation of the above algorithm labels should denote specific transition functions rather than reaction rules. While for this particular example it was sufficient to indicate what "kind" of changes (i.e., reaction rules) need to occur in the system for automated generation of behavior policies it is necessary to distinguish exactly what transformation (including who participated in a specific transformation) is required.

For more detailed examples we refer to [36].

### 2.5. Verification and Visualization of Behavior Policies

Below we will describe the algorithm to verify and illustrate the behavior policy. It consists of 4 phases. At the beginning of the discussion about each phase formal elements not introduced so far will be defined. Subsequent phases will be discussed so that newly introduced definitions will be directly used in the discussed phase. A diagram of relationships between phases is presented in Figure 3, from which it can be seen that the implementation of all the other phases is necessary for the execution of Phase 1. In contrast, Phases 4 and 2 are independent of the others.

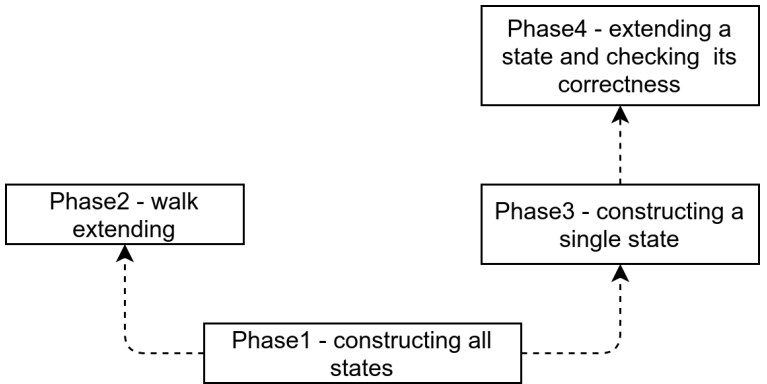

**Figure 3.** Diagram of relationships between phases of the algorithm. The direction of an arrow indicates the phase required by the phase from which the arrow emerges.

### 2.5.1. Phase 4—Applying a Single Transformation to Constructed State and Checking Correctness Beforehand

Phase 4 of the algorithm is responsible for verifying the correctness of the model and for expanding the scenario's state at a particular point in time.

Input:

- A currently constructed state—a bigraph;
- A map of unique identifiers to vertices of the currently constructed state (a bijection);
- The reaction rule to be applied to the constructed state;
- A map of unique identifiers to rule's redex vertices (bijection);
- State at the previous moment in time—a bigraph;
- A mapping of unique identifiers to state vertices at a previous point in time;
- First new unique identifier—used when a new task element appears after a transformation.

Output:

1. Option 1—the model is correct:

   - Newly constructed state—bigraph;
   - Mapping of unique identifiers to vertices of the newly constructed state;
   - First new unique identifier.

2. Option 2—the model is incorrect:

   - Information about the failed transformation. Whether the given reaction rule could not be applied to the state at the previous point in time or to the currently constructed state (given the mappings of unique identifiers to vertices).

Formal definitions:

- $X \subseteq \mathbb{N}$—a set of unique identifiers (UIs) of task elements; It is used to track the environment and objects involved between system transformations. The idea behind this set is to assign to each task element a unique identifier, which makes it possible to check whether the task elements marked as taking part in a reaction rule are present in a given scenario state. The reaction rules themselves allow only to check whether alike (rather than the same) elements exist in both a reaction rule and a bigraph.
- $Corr_{Red} : \mathcal{R} \to Red$—a function that assigns reaction rules to their corresponding redexes;
- $M_x \subset X^{V_b}$　$b \in Agt \cup Red$—a set of functions assigning unique identifiers to elements of the support of a bigraph, which is either a scenario state or a redex of a reaction rule;
- $IsUpdatePossible : Agt \times M_x \times Red \times M_x \to \{true, false\}$—a function that determines whether it is possible to apply a reaction rule to a given state, taking into account the mapping of the UIs to the state's vertices and the mapping of the UIs to the redex vertices of that rule;
- $Update : Agt \times M_x \times \mathcal{R} \times M_x \times X \to Agt \times M_x \times X$—a function that transforms the current state.

The flowchart of the Phase 4 algorithm is shown in Scheme 1. The input arguments of this algorithm and its results are described in Tables 5 and 6 respectively.

**Table 5.** Input data for the Phase 4 algorithm.

| Variable | Description |
| --- | --- |
| $s \in Agt$ | Currently constructed scenario state |
| $m_s \in M$ | Mapping of UIs to vertices of currently constructed state $s$ |
| $r \in \mathcal{R}$ | Reaction rule |
| $m_r \in M_x$ | Mapping of UIs to redex $r$ vertices |
| $s_0 \in Agt$ | State at the previous moment in time |
| $m_0 \in M_x$ | Mapping of UIs to vertices of $s_0$ |
| $n_x \in X$ | The first new UI |

**Table 6.** Output data of the Phase 4 algorithm.

| Variable | Description |
|---|---|
| $res_s \in Agt$ | Constructed state extended by application of the provided reaction rule |
| $res_m \in M_x$ | Mapping of UIs to the vertices of $res_s$ |
| $res_x \in X$ | The first new UI |

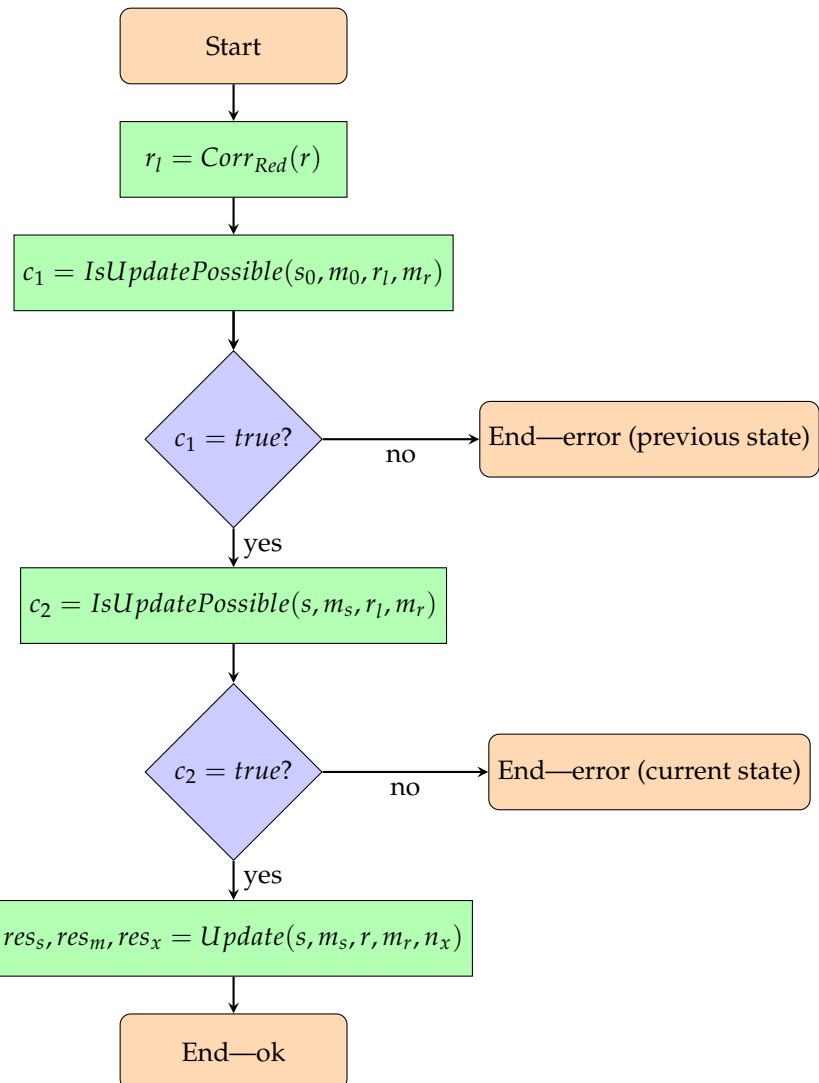

**Scheme 1.** Flowchart of the Phase 4 algorithm. The purpose of this phase is to check if a reaction rule extended by a map of unique identifiers to its vertices can be applied to the scenario state for the previous moment in time and the currently constructed one. If it is impossible to perform either of the mentioned operations it means that the model is incorrectly constructed. If both operations are feasible, the currently constructed state is modified based on the given reaction rule and the map of unique identifiers to its vertices.

### 2.5.2. Phase 3—Constructing Scenario State at a Given Moment of Time

Phase 3 of the algorithm is responsible for constructing the state of the scenario at a given point in time.

Input:

- State at the previous moment in time—bigraph;
- A map of unique identifiers to state elements at the previous moment in time;

- A set of walk elements combined with a UIs mapping to the vertices of the redex of the reaction rule associated with this walk element, a UIs mapping to the vertices of the input state and the smallest new UI from which new task elements will be numbered.
- A linear order relation defined on the above set;
- State-At-Time configuration of the system at the previous moment in time;
- A moment of time for which the system state is constructed;
- Number of objects.

Output:

1. Option 1—the model is correct:

   - A subset of the walk elements (given as input) that have not been used to construct the state at the given point in time;
   - State at the given moment in time;
   - Mapping of unique identifiers to state elements at the given point in time;
   - State-at-time configuration at the set point in time.

2. Option 2—the model is incorrect:

   - A currently constructed state with its UIs mapping that could not be transformed (if it is the cause of the Phase 4 error);
   - The state from the previous moment in time with its UIs mapping that could not be transformed (if it is the cause of the Phase 4 error);
   - Reaction rule with UIs mapping to its redex vertices, which was not successfully applied.

Formal definitions:

- $A \subset 2^{\mathbb{N}}$—A collection of sets of mission object identifiers. The same identifiers are used in SAT configurations
- $W_M \subset \mathbb{N} \times T \times M_x \times M_x \times X \times (A \times \mathbb{N})$—an extended walk consisting of:

  1. A positional number;
  2. A transition function;
  3. A map of UIs to redex vertices. The redex is associated with the reaction rule corresponding to the above transition function;
  4. A map of UIs to vertices of the output state of the extended walk element;
  5. First new UI assigned to a new task element created by applying the reaction rule (useful only if the reaction rule corresponding to the transition function creates new environment elements);
  6. A set of object identifiers involved in the walk element along with the duration of that transformation. In other words, it is information about which objects are involved in the transformation represented by the walk element and how long it will take.

- $<_{W_M}$—linear order relation on the elements of the extended walk.
  We will assume the following rule for ordering the elements of a walk:

$$\forall e_1 = (l_1, f_1, m_{1,r}, m_{1,in}, n_1, (A_1, d_1)), e_2 = (l_2, f_2, m_{2,r}, m_{2,in}, n_2, (A_2, d_2)) \in W_M$$
$$e_1 <_{W_M} e_2 \leftrightarrow l_1 < l_2$$

- $First_M : 2^{W_M} \to W_M \times 2^{W_M}$—a function that returns the "smallest" element of the walk and the "truncated" walk;
- $Corr_{Tra} : T \to Tra$—a function that assigns transition functions to transitions from TTS;
- $Objects_U : I \times (A \times \mathbb{N}) \to I$—SAT configuration update function. Takes a current configuration and a set of objects for which the time will be changed along with the value by how much. The result is the new SAT configuration;
- $Objects_F : I \times \mathbb{N} \to A$—a function that determines for which objects activities are scheduled later than the moment of time for which the scenario state is constructed. Takes a SAT configuration and the moment of time for which the state is generated;

- $Corr_\mathcal{R} : Tra \to \mathcal{R}$—a function assigning reaction rules to transitions from TTS.

    The flowchart of Phase 3 of the algorithm is shown in Scheme 2. The input arguments for the algorithm are described in Table 7. The auxiliary variables, some of which are also outcomes of Phase 3, are described in Table 8. The outcome of Phase 3 is described in Table 9.

**Table 7.** Input data for the Phase 3 algorithm.

| Variable | Description |
|---|---|
| $s_0 \in Agt$ | State at the previous point in time. |
| $m_0 \in M_x$ | Mapping of UIs to vertices of $s_0$. |
| $W \subseteq W_M, <_{W_M}$ | A walk and the linear order relation on its elements. |
| $i_0 \in I$ | The SAT configuration at the previous moment of time. |
| $d \in \mathbb{N}$ | The moment of time for which the scenario state is constructed. |
| $n_o \in \mathbb{N}$ | Number of objects. |

**Table 8.** Auxiliary variables of Phase 3 algorithm.

| Variable | Description |
|---|---|
| $s_c \in Agt$ | Current constructed state. The initial value is $s_0$. |
| $m_c \in M_x$ | Mapping of UIs to vertices of $s_c$. |
| $i_c \in I$ | SAT configuration of the currently constructed state. The initial value is $i_0$. |
| $A_o \in A$ | A set of object identifiers, skipped in the constructed state. The initial value is the empty set. |
| $W_c \subseteq W$ | A collection of usable walk elements. The initial value is $W$. |
| $W_o \subseteq W$ | A collection of unused walk elements. The initial value is the empty set. |

**Table 9.** Output data of Phase 3 algorithm.

| Variable | Description |
|---|---|
| $W_o \subset W$ | Unused walk elements that will be used to construct subsequent scenario states. |
| $s_c \in Agt$ | System state. |
| $m_c \in M_x$ | Mapping of UIs to vertices of $s_c$. |
| $i_c \in I$ | SAT configuration at time $d$. |

Noteworthy are the conditions checked in the subsequent steps of Phase 3 of the algorithm. Comments for each of them are given below.

1. The first condition checked is if we have reached the end of a walk. If so, then surely the state currently constructed is the state for the given moment of time.
2. Do we omit actions of all mission objects? If so, the state constructed so far is the state for the given moment of time.
3. Do any objects involved in the current action belong to the set of skipped objects? If so, we omit this walk element.
4. Will all objects involved in the current action have finished before the moment $d$? If not, we disregard that activity in the currently constructed state and add those objects to the set of skipped objects.
5. If Phase 4 is not completed correctly, it means that the model is incorrect.

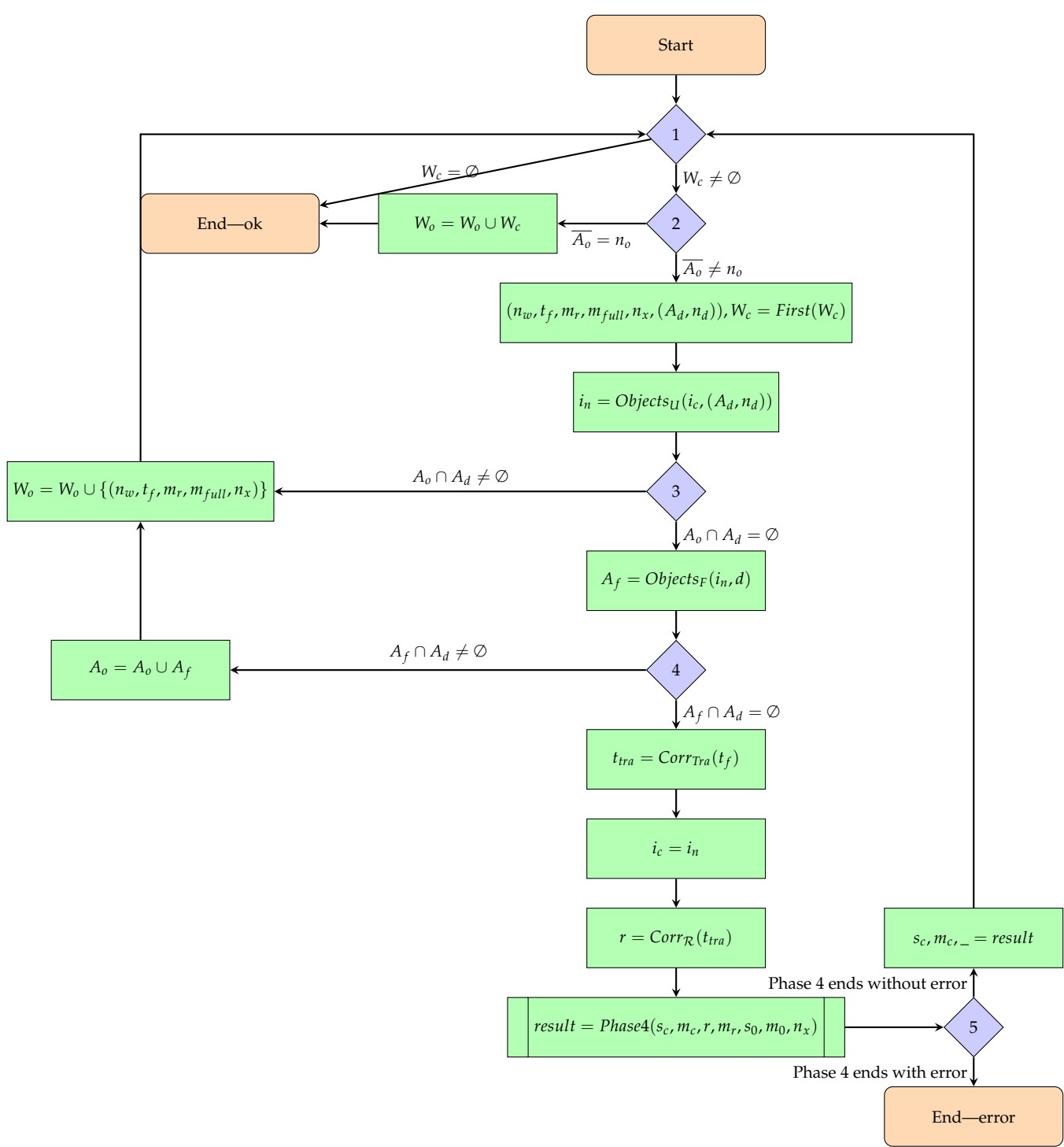

**Scheme 2.** Flowchart of the Phase 3 algorithm. The goal of this phase is to construct the state of a scenario at a given point in time. This phase runs in a loop until there are no available walk elements or when an execution of Phase 4 ends with an error. It takes subsequent elements of the input walk and updates both the current SAT configuration and a scenario state. If the mission objects will not have finished the activity represented by the currently processed walk element before or at the specified moment of time then the SAT configuration and state updates are not performed. The same thing happens if an activity involves objects participating in other activities that would end in a future and that have already been skipped.

### 2.5.3. Phase 2—Extending a Previously Constructed Walk

Phase 2 of the algorithm is responsible for extending a walk to the form acceptable by Phase 3.

Input:

- A walk resulting from the algorithm presented in Section 2.4;
- Number of objects.

Output:

- Extended walk.

Formal definitions:

- $W \subset \mathbb{N} \times T$—a walk. The first element denotes the positional number of the transition function that is the second element of the tuple;
- $<_W$—linear order relation on the elements of the set $W$.
  As in the case of the set $W_M$, we define the order relation by the following rule:

$$\forall e_1 = (n_1, f_1), e_2 = (n_2, f_2) \in W \quad e_1 <_W e_2 \leftrightarrow n_1 < n_2$$

- $First : 2^W \to W \times 2^W$—a function that returns the "smallest" walk element and a truncated walk;
- $Trans : Tra \times M_x \times X \to M_x \times M_x \times X$—a function that transforms a mapping of unique identifiers based on the given transition and the first new identifier (in case new environment elements appear in the output state of the transition and need to be tagged). The results are: a new UIs map to the redex of the reaction rule corresponding to the provided transition, a UIs mapping to the output state of the transition, and a new smallest UI;
- $U \subset I^I$—a set of functions that update SAT configurations;
- $Corr_U : T \to U$—a function that assigns transition functions to their corresponding SAT configuration update functions;
- $Time_U : I \times U \to I$—a SAT configuration update function;
- $Time_D : I \times I \to A \times \mathbb{N}$—a time difference function for individual objects between SAT configurations. Returns information about which objects are involved in the transformation and how long it takes.

The flowchart of the Phase 2 algorithm is shown in Scheme 3. Input arguments are described in Table 10; auxiliary variables and the result of this phase are discussed in Table 11.

**Table 10.** Input data for the Phase 2 algorithm.

| Variable | Description |
| --- | --- |
| $W, <_W$ | A walk with a linear order relation on its elements. |
| $n_o$ | Number of objects. |

**Table 11.** Auxiliary variables of the Phase 2 algorithm.

| Variable | Description |
| --- | --- |
| $n_x \in X$ | The value of a first new UI. The initial value is the number of vertices of the input state of the first walk element. |
| $m_{full} \in M_x$ | The current UIs mapping to the vertices of the last processed output state. The initial value is a function that assigns consecutive natural numbers to the vertices of the input state of the first element of the walk. |
| $W_r \subseteq W_M$ | Elements of the extended walk. The initial value is the empty set. This is the result of this phase. |
| $W_c \subseteq W$ | A subset of walk elements that have not been processed yet. The initial value is $W$. |
| $i_c \in I$ | Current SAT configuration. The initial value is $((1, 0), \ldots, (n_o, 0))$. |

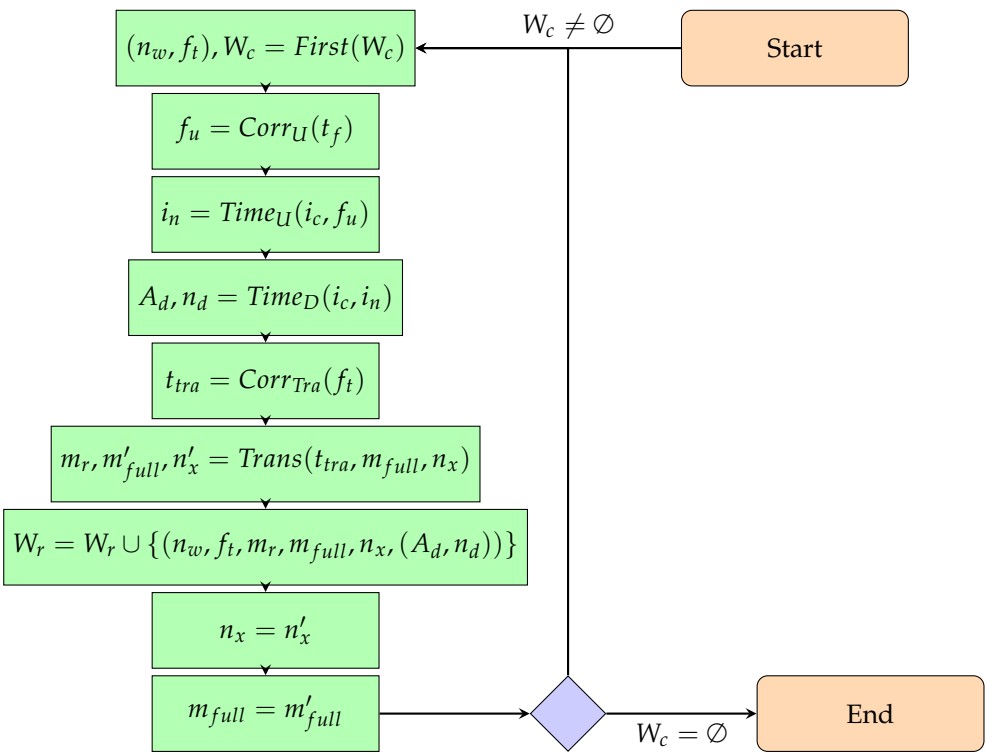

**Scheme 3.** Flowchart of the Phase 2 algorithm. The goal of Phase 2 is to expand each element of a provided walk to the form acceptable by Phase 3. Each element of the walk is coupled with the duration of its corresponding activity along with the identifiers of the objects (not unique identifiers of task elements) that participate in the activity and two bijections. The first function maps unique identifiers to vertices of the redex of the reaction rule associated with the currently processed walk element. With this function, we know exactly who is participating in the activity. The second function maps unique identifiers to the output state of a processed TTS transition (derived from the walk element). With this function, we know exactly which task element corresponds to which vertex after applying the reaction rule. The second function is used in the next iteration of Phase 2.

### 2.5.4. Phase 1—Constructing All Scenario States and Checking the Correctness of a Given Walk

Phase 1 of the algorithm is its entry point. It is responsible for verifying a model and constructing scenario states at successive moments in time.

Input:

- Number of objects;
- A walk with a linear order relation on its elements.

Output:

1. The model is correct:

    - A set of scenario states at consecutive moments in time with corresponding mappings of unique identifiers to the vertices of these states and SAT configurations;

2. The model is incorrect:

    - The moment of time for which the scenario state could not be generated;
    - The element that could not be transformed (constructed state or state at some point in time);
    - The reaction rule corresponding to the unsuccessful transformation;
    - The UIs mapping of the element that could not be transformed and the redex of the above reaction rule.

Phase 1 input parameters are described in Table 12. The auxiliary variables along with the outcome are discussed in Table 13. The flowchart of the Phase 1 algorithm is shown in the Scheme 4.

**Table 12.** Input data for the Phase 1 algorithm.

| Variable | Description |
|---|---|
| $W, <_W$ | A walk with a linear order relation on its elements. |
| $n_o$ | Number of objects. |

**Table 13.** Auxiliary variables for the Phase 1 algorithm.

| Variable | Description |
|---|---|
| $W_c \subseteq W_M$ | A set of extended walk elements that have not been used yet. The initial value is the empty set but it is properly initialized with the result of Phase 2. |
| $d$ | The current moment of time for which a scenario state is constructed. The initial value is 1. |
| $s \in Agt$ | The scenario state at the time $d - 1$. The initial value is the input state of the first walk element $W$. |
| $m_s \in M_x$ | Mapping of UIs to vertices of the state $s$. The initial value is a bijection of consecutive natural numbers on the vertices of $s$. |
| $i_s \in I$ | SAT configuration for the scenario state at the time $d - 1$. The initial value is $((1,0), \ldots, (n_o, 0))$. |
| $S_r \subset \mathbb{N} \times Agt \times M_x \times I$ | A collection of states at successive moments in time with their corresponding UIs mapping and SAT configurations. The initial value is the empty set. This is the result of this phase. |

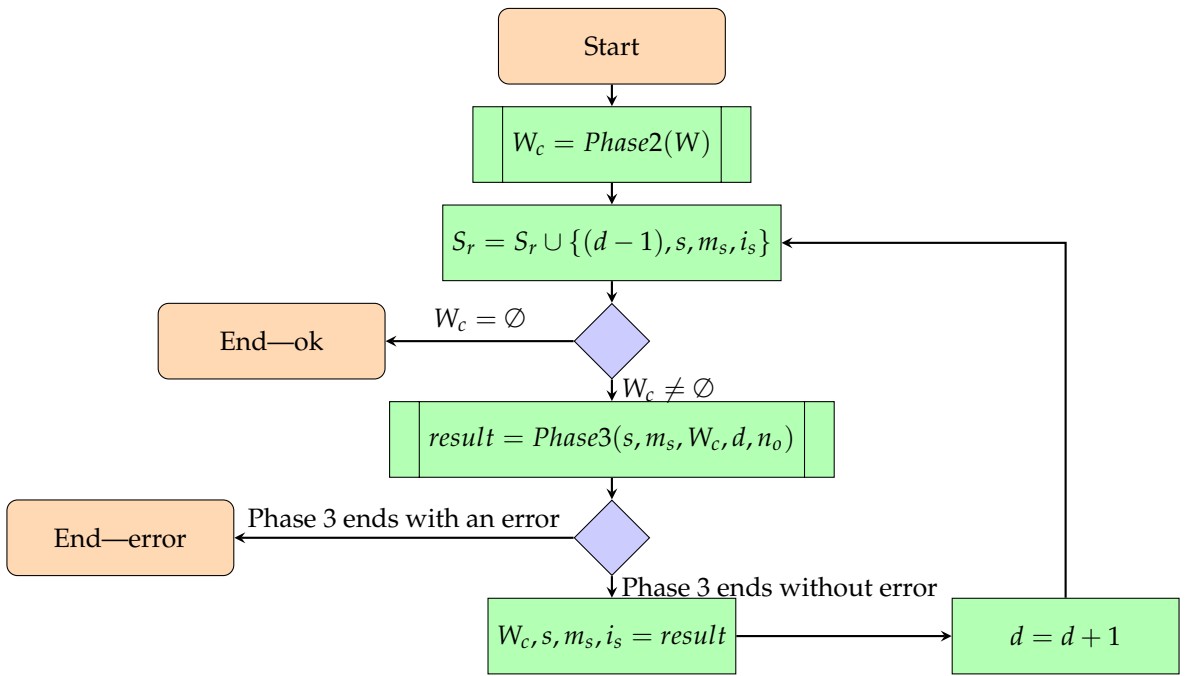

**Scheme 4.** A flowchart of the Phase 1 algorithm. The goal of Phase 1 is to verify a model and construct the subsequent states of a scenario using a provided walk. In the first step the walk is extended to the form acceptable by Phase 3. Then the model verification and construction of successive scenario states is performed in a loop. The loop is executed until Phase 3 ends with either an error or when there are no more elements of the walk to further construct states of the scenario from.

## 3. Results

This section will provide example use cases of the algorithm discussed in the previous section. The first two examples show in detail how the algorithm detects errors in a model and how it constructs successive scenario states. The next examples present how to check the fulfillment of non-functional requirements for systems designed with our methodology. Finally, the problem of memory complexity of convolution operation performed during a construction of walks in a state space is discussed. We also provide a few propositions how to address this issue.

### 3.1. Model Verification Example

#### 3.1.1. Introduction

The first example will demonstrate how the algorithm can detect that a system is incorrectly designed.

A task (as defined in Section 2.1) for this example consists of six elements, two actions that can be performed, and one goal. The task elements comprise three areas with two robots and an object to be carried between the areas. The goal of the task is for the robots (denoted by vertices with the control *B*) to move the object (denoted by a vertex with the control *O*) from the area *AT1* to the area *AT3*. The initial state of this system is shown in Figure 4. We will use two reaction rules to generate a tracking bigraphical reactive system: *mov1* and *mov2* depicted in Figure 5a,b, respectively.

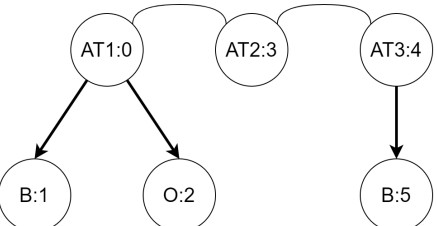

**Figure 4.** The initial state of a system in the example of verifying model correctness.

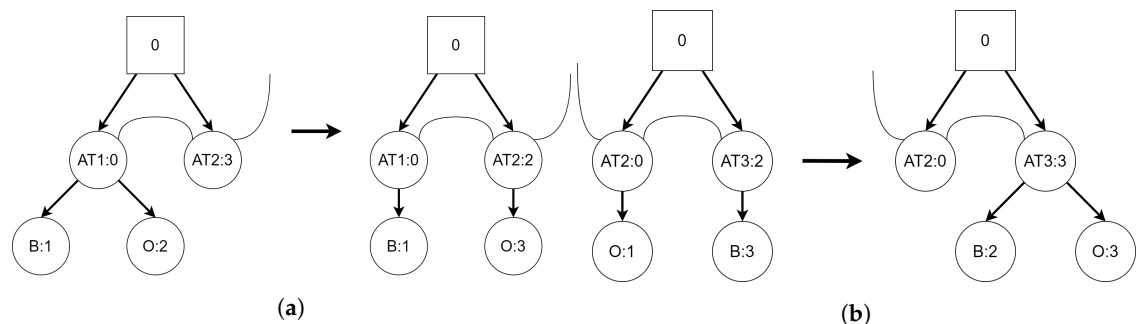

(**a**)　　　　　　　　　　　　　(**b**)

**Figure 5.** Reaction rules for the example of verifying a model. All residue functions are identities. (**a**) Reaction rule *mov*1. (**b**) Reaction rule *mov*2.

The elements of a tracking transition system for this example are shown in Table 14.

If we categorize the task elements as presented in Table 15 then we can transform the TTS from Table 14 into the state space as in Figure 6. However, this will not be a valid state space because no time is taken into account for the object being moved (i.e., it is not treated as a passive or active object as defined in Section 2.1).

#### 3.1.2. Using the Algorithm for Model Verification

Walk $S_0 \xrightarrow{f_1} S_1 \xrightarrow{f_2} S_2$ can be represented as $W = \{(0, f_1), (1, f_2)\}$. Assuming that both actions associated to the reaction rules take 1 unit of time to complete, in Phase 2 both elements of set $W$ will be transformed to form:

1.　$(0, f_1, \{(0,0),(1,1),(2,2),(3,3)\}, \{(0,0),(1,1),(2,2),(3,3),(4,4),(5,5)\}, 6, (\{1\},1))$
2.　$(1, f_2, \{(2,0),(3,1),(4,2),(5,3)\}, \{(0,0),(1,1),(2,3),(3,2),(4,4),(5,5)\}, 6, (\{2\},1))$

The method of constructing $m_r$ and $m'_{full}$ functions that result from *Trans* function in Phase 2 is shown below.

The rule of constructing $m_r$ function:

$$\forall x \in \{0, \dots, n_x - - - 1\} \quad m_r(x) = par^{-1}(m_{full}(x))$$

where $par^{-1}$ is the inverse function to *par* being an element of $t_{tra}$. In this case, the functions $f_1, f_2, \dots, f_8$ correspond to the subsequent rows in Table 14.

The rule of constructing $m'_{full}$ function:

$$\forall x \in \{0, \dots, n_x - - - 1\} \quad m'_{full}(x) = res^{-1}(m_{full}(x))$$

$res^{-1}$ is the inverse function of *res* which is an element of $t_{tra}$.

Table 16 lists the successive steps of the algorithm that will lead to a detection of an error in the model. The reason why this model is incorrect is not because the redex of the rule *mov2* is not in the *0* state but because the moved object is categorized as an element of the environment, thus we do not take into account the passage of time for it. As a result, the reaction rules create the appearance of being independent of each other when in fact the execution of *mov2* rule is dependent on the execution of the rule *mov1*. To fix the model, the relocated object needs to be categorized as a passive object and one need to add a reaction rule allowing a robot that is in *AT3* area to wait until the object being moved is in *AT2* area.

**Table 14.** Tracking transition system for the first example.

| Input State | Label | Output State | Par & Res |
|---|---|---|---|
| | *mov*1 | | $\{(0,0),(1,1),(2,2),(3,3)\}$ $\{(0,0),(1,1),(2,3),(3,2),(4,4),(5,5)\}$ |
| | *mov*2 | | $\{(0,2),(1,3),(2,4),(3,5)\}$ $\{(0,2),(1,4),(2,5),(3,3),(4,0),(5,1)\}$ |

**Table 15.** Categorization of task elements for the first example. Note that this produces an incorrect model because the moved object is considered an environment element.

| Category of Task Elements | Elements Belonging to the Category |
|---|---|
| Environment | $\{AT1, AT2, AT3, O\}$ |
| Passive objects | $\varnothing$ |
| Active objects (agents) | $\{B\}$ |

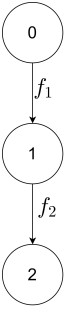

**Figure 6.** Incorrect state space for the task from the first example.

**Table 16.** Subsequent steps of the algorithm in the model validation example.

| Phase | Step | Result/Comment |
|---|---|---|
| 1 | $S_r = S_r \cup \{((d-1, s, m_s, i_s))\}$ | $S_r = \left\{ \left( \begin{array}{c} 0, \text{(bigraph: AT1:0, AT2:3, AT3:4; B:1, O:2, B:5)} , \\ \{(0,0), (1,1), (2,2), (3,3), (4,4), (5,5)\}, \\ ((1,0), (2,0)) \end{array} \right) \right\}$ |
| 1 | $Phase3(\dots)$ | |
| 3 | $n_w, t_f, m_r, m_{full}, n_x, (A_d, n_d), W_c = First_M(W_c)$ | $n_w = 0$ <br> $t_f = f_1$ <br> $m_r = \{(0,0), (1,1), (2,2), (3,3)\}$ <br> $m_{full} = \{(0,0), (1,1), (2,2), (3,3), (4,4), (5,5)\}$ <br> $n_x = 6$ <br> $A_d = \{1\}$ <br> $n_d = 1$ <br> $W_c = W_c \setminus \{e_1\} = \{e_2\}$ |
| 3 | $i_n = Objects_U(i_c, (A_d, n_d))$ | $i_n = ((1,1), (2,0))$ |
| 3 | $A_f = Objects_F(i_n, d)$ | $A_f = \varnothing$ |
| 3 | $t_{tra} = Corr_{Tra}(t_f)$ | $t_{tra} = $ first row of Table 14 |
| 3 | $i_c = i_n$ | $i_c = (1,1), (2,0)$ |
| 3 | $r = Corr_R(t_{tra})$ | $r = $ reaction rule *mov1* |
| 3 | $result = Phase4(s_c, m_c, r, s_0, m_0, n_x)$ | Phase 4 completed without error |
| 3 | $s_c, m_c, n_x = result$ | $s_c = $ (bigraph: AT1:0→B:1, AT2:2→O:3, AT3:4→B:5) <br><br> $m_c = \{(0,0), (1,1), (2,3), (3,2), (4,4), (5,5)\}$ <br> ($m_c$ is calculated in the same way as $m'_{full}$ in Phase 2) <br> $n_x = 6$ |
| 3 | $n_w, t_f, m_r, m_{full}, n_x, (A_d, n_d), W_c = First_M(W_c)$ | $n_w = 1$ <br> $t_f = f_2$ <br> $m_r = \{(2,0), (3,1), (4,2), (5,3)\}$ <br> $m_{full} = \{(0,0), (1,1), (2,3), (3,2), (4,4), (5,5)\}$ <br> $n_x = 6$ <br> $A_d = \{2\}$ <br> $n_d = 1$ <br> $W_c = W_c \setminus \{e_2\} = \varnothing$ |
| 3 | $i_n = Objects_U(i_c, (A_d, n_d))$ | $i_n = ((1,1), (2,1))$ |
| 3 | $A_f = Objects_F(i_n, d)$ | $A_f = \varnothing$ |
| 3 | $t_{tra} = Corr_{Tra}(t_f)$ | $t_{tra} = $ second row of Table 14 |
| 3 | $i_c = i_n$ | $i_c = ((1,1), (2,1))$ |
| 3 | $r = Corr_R(t_{tra})$ | $r = $ reaction rule *mov2* |
| 4 | $r_l = Corr_{Red}(r)$ | $r_l = $ (bigraph: 0 → AT2:0, AT3:2; O:1, B:3) |
| 4 | $c_1 = IsUpdatePossible(s_0, m_0, r_l, m_r)$ | $c_1 = false$ <br><br> The pattern (bigraph: 0 → AT2:0, AT3:2; O:1, B:3) does not occur in the bigraph (AT1:0, AT2:3, AT3:4; B:1, O:2, B:5). |
| 1 | End—error | |

### 3.2. Example of Scenario States Visualization

#### 3.2.1. Introduction

The second example will demonstrate the problem of visualizing a scenario and how our algorithm can help in solving it. A task for this example is composed of three areas and two robots of the same type. The initial state of the system is presented in Figure 7. The tracking bigraphical reactive system for the purpose of this example consists of two reaction rules, *r1* and *r2*, shown in Figure 8a,b, respectively. The goal of the task is to move the two robots from the area *AT1* to the area *AT3*.

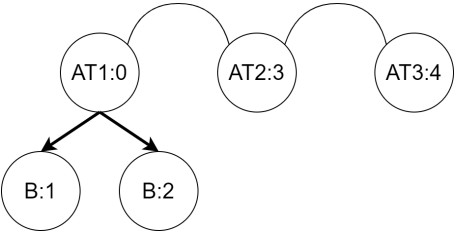

**Figure 7.** The initial state of a system for the scenario visualization example.

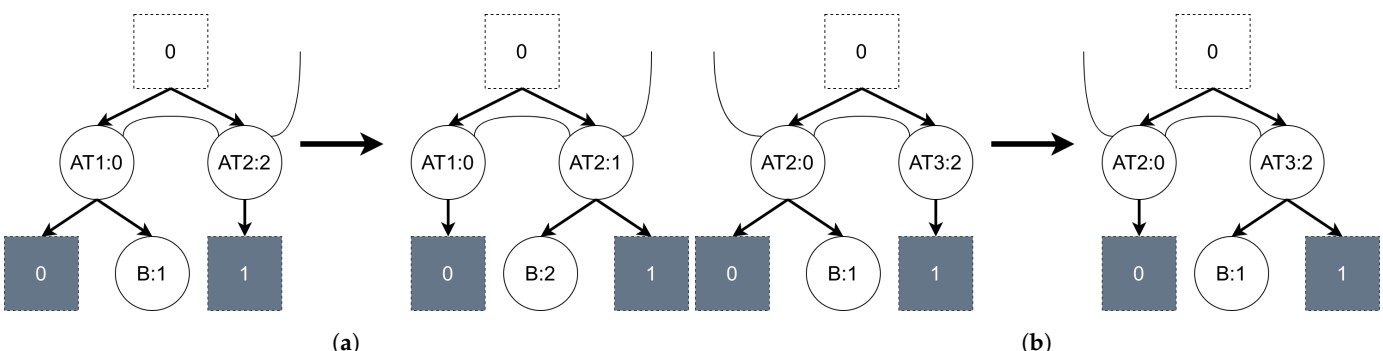

**(a)**                                                           **(b)**

**Figure 8.** Reaction rules for the example of scenario visualization. (**a**) Reaction rule $r1$. $\tau = \{(0,0),(1,2),(2,1)\}$. (**b**) Reaction rule $r2.\tau = \{(0,0),(1,1),(2,2)\}$.

Tracking Transition System generated from this TBRS is defined in Table 17.

The tracking Transition System from Table 17 can be transformed into a state space as in Figure 9. Now, suppose that a walk chosen for the behavior policy is of the form:

$$S_0 \xrightarrow{f_1} S_1 \xrightarrow{f_3} S_2 \xrightarrow{f_5} S_4 \xrightarrow{f_8} S_5$$

The above walk never "passes" through a state where both robots are in *AT2* area. (that is, through the state $S_3$). Such a situation must occur for the following reasons. For a walk representing four activities (because it consists of four arcs), that can correspond only to reaction rules from Figure 8 a course of a mission for each robot must take the form of moving from an AT1 area to an AT2 area and then from AT2 to AT3 area. Since the activities represented by the reaction rules are not cooperative (each of the reaction rules involve only one agent) the movements will be performed in parallel. We also know that the time required to perform both activities will be the same for both agents (because agents are of the same type and perform the same type of activity) so the successive movements will end at the same moment. Because of all that, during a mission there must occur a situation where both robots are at an AT2 area at the same time. Therefore, the algorithm for constructing subsequent scenario states must be able to construct states that are not "on" a provided walk.

**Table 17.** Tracking Transition System for the second example.

| Input State | Label | Output State | Par & Res |
|---|---|---|---|
| (AT1:0, AT2:3, AT3:4 / B:1, B:2) | $r1$ | (AT1:0, AT2:1, AT3:4 / B:3, B:2) | $par = \{(0,0),(1,2),(2,3)\}$ $res = \{(0,0),(1,3),(2,2),(3,1),(4,4)\}$ |
| (AT1:0, AT2:3, AT3:4 / B:1, B:2) | $r1$ | (AT1:0, AT2:1, AT3:4 / B:3, B:2) | $par = \{(0,0),(1,1),(2,3)\}$ $res = \{(0,0),(1,3),(2,1),(3,2),(4,4)\}$ |
| (AT1:0, AT2:1, AT3:4 / B:3, B:2) | $r2$ | (AT1:3, AT2:0, AT3:1 / B:4, B:2) | $par = \{(0,1),(1,2),(2,4)\}$ $res = \{(0,1),(1,4),(2,2),(3,0),(4,3)\}$ |
| (AT1:0, AT2:1, AT3:4 / B:3, B:2) | $r1$ | (AT1:0, AT2:1, AT3:4 / B:3, B:2) | $par = \{(0,1),(1,3),(2,1)\}$ $res = \{(0,0),(1,1),(2,2),(3,3),(4,4)\}$ |
| (AT1:3, AT2:0, AT3:1 / B:4, B:2) | $r1$ | (AT1:0, AT2:1, AT3:3 / B:2, B:4) | $par = \{(0,3),(1,4),(2,0)\}$ $res = \{(0,3),(1,0),(2,4),(3,1),(4,2)\}$ |
| (AT1:0, AT2:1, AT3:4 / B:3, B:2) | $r2$ | (AT1:0, AT2:1, AT3:3 / B:2, B:4) | $par = \{(0,1),(1,2),(2,4)\}$ $res = \{(0,0),(1,1),(2,3),(3,4),(4,2)\}$ |
| (AT1:0, AT2:1, AT3:4 / B:3, B:2) | $r2$ | (AT1:0, AT2:1, AT3:3 / B:2, B:4) | $par = \{(0,1),(1,3),(2,4)\}$ $res = \{(0,0),(1,1),(2,2),(3,4),(4,3)\}$ |
| (AT1:0, AT2:1, AT3:3 / B:2, B:4) | $r2$ | (AT1:3, AT2:0, AT3:1 / B:2, B:4) | $par = \{(0,1),(1,2),(2,3)\}$ $res = \{(0,0),(1,3),(2,4),(3,0),(4,2)\}$ |

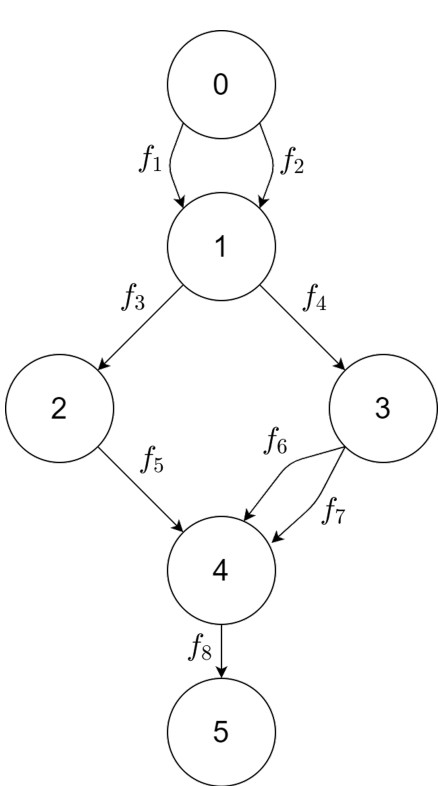

**Figure 9.** The state space generated from Tracking Transition System from Table 17.

3.2.2. Using the Algorithm to Construct Scenario States

The walk $S_0 \xrightarrow{f_1} S_1 \xrightarrow{f_3} S_2 \xrightarrow{f_5} S_4 \xrightarrow{f_8} S_5$ can be presented as:

$$W = \{(0, f_1), (1, f_3), (2, f_5), (3, f_8)\}$$

A linear order relation on the set $W$ has the form:

$$<_W = \begin{Bmatrix} ((0, f_1), (1, f_3)), ((0, f_1), (2, f_5)), ((0, f_1), (3, f_8)), \\ ((1, f_3), (2, f_5)), ((1, f_3), (3, f_8)), \\ ((2, f_5), (3, f_8)) \end{Bmatrix}$$

Assuming that execution of each reaction rules takes one unit of time, in Phase 2 the consecutive elements of set $W$ will be transformed to the following form:

- $e_1 = (0, f_1, \{(0,0), (2,1), (3,2)\}, \{(0,0), (1,1), (2,2), (3,3), (4,4)\}, 5, (\{2\}, 1))$
- $e_2 = (1, f_3, \{(3,0), (2,1), (4,2)\}, \{(0,0), (3,1), (2,2), (1,3), (4,4)\}, 5, (\{2\}, 1))$
- $e_3 = (2, f_5, \{(0,0), (1,1), (3,2)\}, \{(0,3), (1,4), (2,2), (3,0), (4,1)\}, 5, (\{1\}, 1))$
- $e_4 = (3, f_8, \{(3,0), (1,1), (4,2)\}, \{(0,0), (1,2), (2,4), (3,1), (4,3)\}, 5, (\{1\}, 1))$

Knowing the above, we can define an extended walk.

$$W_M = \{e_1, e_2, e_3, e_4\}$$

The linear order relation remains unchanged between elements, i.e,:

$$<_{W_M} = \{(e_1, e_2), (e_1, e_3), (e_1, e_4), (e_2, e_3), (e_2, e_4), (e_3, e_4)\}$$

Steps of the algorithm to construct the subsequent scenario states are presented in Table 18.

**Table 18.** Successive steps of the algorithm in the example of visualizing a scenario.

| Phase | Step | Result/Comment |
|:---:|:---:|:---|
| 1 | $S_r = S_r \cup \{((d-1, s, m_s, i_s))\}$ | $S_r = \left\{ \begin{pmatrix} 0, \text{} , \\ \{(0,0), (1,1), (2,2), (3,3), (4,4)\}, \\ ((1,0), (2,0)) \end{pmatrix} \right\}$ |
| 1 | $Phase3(\dots)$ | |
| 3 | $n_w, t_f, m_r, m_{full}, n_x, (A_d, n_d), W_c = First_M(W_c)$ | $n_w = 0$ <br> $t_f = f_1$ <br> $m_r = \{(0,0), (2,1), (3,2)\}$ <br> $m_{full} = \{(0,0), (3,1), (2,2), (1,3), (4,4)\}$ <br> $n_x = 5$ <br> $A_d = \{2\}$ <br> $n_d = 1$ <br> $W_c \quad = $ <br> $W_c \setminus \{e_1\} \quad = $ <br> $\{e_2, e_3, e_4\}$ |
| 3 | $i_n = Objects_U(i_c, (A_d, n_d))$ | $i_n = ((1,0), (2,1))$ |
| 3 | $A_f = Objects_F(i_n, d)$ | $A_f = \varnothing$ |
| 3 | $t_{tra} = Corr_{Tra}(t_f)$ | $t_{tra} = $ first row of Table 17 |
| 3 | $i_c = i_n$ | $i_c = ((1,0), (2,1))$ |
| 3 | $r = Corr_\mathcal{R}(t_{tra})$ | $r = $ reaction rule $r1$ |
| 3 | $result = Phase4(s_c, m_c, r, s_0, m_0, n_x)$ | Phase 4 completed without error |

**Table 18.** *Cont.*

| Phase | Step | Result/Comment |
|---|---|---|
| 3 | $s_c, m_c, n_x = result$ | $s_c =$  |
| | | $m_c = \{(0,0),(3,1),(2,2),(1,3),(4,4)\}$ |
| | | $n_x = 5$ |
| 3 | $n_w, t_f, m_r, m_{full}, n_x, (A_d, n_d), W_c = First_M(W_c)$ | $n_w = 1$ |
| | | $t_f = f_3$ |
| | | $m_r = \{(3,0),(2,1),(4,2)\}$ |
| | | $m_{full} = \{(0,0),(3,1),(2,2),(1,3),(4,4)\}$ |
| | | $n_x = 5$ |
| | | $A_d = \{2\}$ |
| | | $n_d = 1$ |
| | | $W_c = W_c \setminus \{e_2\} = \{e_3, e_4\}$ |
| 3 | $i_n = Objects_U(i_c, (A_d, n_d))$ | $i_n = ((1,0),(2,2))$ |
| 3 | $A_f = Objects_F(i_n, d)$ | $A_f = \{2\}$ |
| 3 | $A_o = A_o \cup A_f$ | $A_o = \varnothing \cup \{2\} = \{2\}$ |
| 3 | $W_o = W_o \cup \{(n_w, t_f, m_r, m_{full}, n_x, (A_d, n_d))\}$ | $W_o = \varnothing \cup \{e_2\}$ |
| 3 | $n_w, t_f, m_r, m_{full}, n_x, (A_d, n_d), W_c = First_M(W_c)$ | $n_w = 2$ |
| | | $t_f = f_5$ |
| | | $m_r = \{(0,0),(1,1),(3,2)\}$ |
| | | $m_{full} = \{(0,3),(1,4),(2,2),(3,0),(4,1)\}$ |
| | | $n_x = 5$ |
| | | $A_d = \{1\}$ |
| | | $n_d = 1$ |
| | | $W_c = W_c \setminus \{e_3\} = \{e_4\}$ |
| 3 | $i_n = Objects_U(i_c, (A_d, n_d))$ | $i_n = ((1,1),(2,1))$ |
| 3 | $A_f = Objects_F(i_n, d)$ | $A_f = \varnothing$ |
| 3 | $t_{tra} = Corr_{Tra}(t_f)$ | $t_{tra} =$ fifth row of Table 17 |
| 3 | $i_c = i_n$ | $i_c = ((1,1),(2,1))$ |
| 3 | $r = Corr_{\mathcal{R}}(t_{tra})$ | $r =$ reaction rule *r1* |
| 3 | $result = Phase4(s_c, m_c, r, s_0, m_0, n_x)$ | Phase 4 completed without error |
| 3 | $s_c, m_c, n_x = result$ | $s_c =$  |
| | | $m_c = \{(0,0),(3,1),(2,2),(1,3),(4,4)\}$ |
| | | $n_x = 5$ |
| 3 | $n_w, t_f, m_r, m_{full}, n_x, (A_d, n_d), W_c = First_M(W_c)$ | $n_w = 3$ |
| | | $t_f = f_8$ |
| | | $m_r = \{(3,0),(1,1),(4,2)\}$ |
| | | $m_{full} = \{(0,0),(1,2),(2,4),(3,1),(4,3)\}$ |
| | | $n_x = 5$ |
| | | $A_d = \{1\}$ |
| | | $n_d = 1$ |
| | | $W_c = W_c \setminus \{e_4\} = \varnothing$ |
| 3 | $i_n = Objects_U(i_c, (A_d, n_d))$ | $i_n = ((1,2),(2,1))$ |
| 3 | $A_f = Objects_F(i_n, d)$ | $A_f = \{1\}$ |

**Table 18.** *Cont.*

| Phase | Step | Result/Comment |
|---|---|---|
| 3 | $A_o = A_o \cup A_f$ | $A_o = \{2\} \cup \{1\} = \{1, 2\}$ |
| 3 | $W_o = W_o \cup \{(n_w, t_f, m_r, m_{full}, n_x, (A_d, n_d))\}$ | $W_o = \{e_2\} \cup \{e_4\}$ |
| 3 | End—ok | |
| 1 | $W_c, s, m_s, i_s = result$ | $W_c = \{e_2, e_4\}$ <br><br> $s =$  <br><br> $m_s = \{(0,0), (3,1), (2,2), (1,3), (4,4)\}$ <br> $i_s = ((1,1), (2,1))$ |
| 1 | $d = d + 1$ | $d = 2$ |
| 1 | $S_r = S_r \cup \{((d-1, s, m_s, i_s))\}$ | $S_r = S_r \cup \left\{ \left( 1, \begin{array}{c} \text{} \\ \{(0,0),(3,1),(2,2),(1,3),(4,4)\}, \\ ((1,1),(2,1)) \end{array} \right) \right\}$ |
| 1 | $Phase3(\ldots)$ | |
| 3 | $n_w, t_f, m_r, m_{full}, n_x, (A_d, n_d), W_c = First_M(W_c)$ | $n_w = 1$ <br> $t_f = f_3$ <br> $m_r = \{(3,0), (2,1), (4,2)\}$ <br> $m_{full} = \{(0,0), (3,1), (2,2), (1,3), (4,4)\}$ <br> $n_x = 5$ <br> $A_d = \{2\}$ <br> $n_d = 1$ <br> $W_c = W_c \setminus \{e_2\} = \{e_4\}$ |
| 3 | $i_n = Objects_U(i_c, (A_d, n_d))$ | $i_n = ((1,1), (2,2))$ |
| 3 | $A_f = Objects_F(i_n, d)$ | $A_f = \varnothing$ |
| 3 | $t_{tra} = Corr_{Tra}(t_f)$ | $t_{tra} = $ third row of Table 17 |
| 3 | $i_c = i_n$ | $i_c = ((1,1), (2,2))$ |
| 3 | $r = Corr_{\mathcal{R}}(t_{tra})$ | $r = $ reaction rule $r2$ |
| 3 | $result = Phase4(s_c, m_c, r, s_0, m_0, n_x)$ | Phase 4 completed without error |
| 3 | $s_c, m_c, n_x = result$ | $s_c =$  <br><br> $m_c = \{(0,0), (3,1), (4,3), (1,2), (2,4)\}$ <br> $n_x = 5$ |
| 3 | $n_w, t_f, m_r, m_{full}, n_x, (A_d, n_d), W_c = First_M(W_c)$ | $n_w = 3$ <br> $t_f = f_8$ <br> $m_r = \{(3,0), (1,1), (4,2)\}$ <br> $m_{full} = \{(0,0), (1,2), (2,4), (3,1), (4,3)\}$ <br> $n_x = 5$ <br> $A_d = \{1\}$ <br> $n_d = 1$ <br> $W_c = W_c \setminus \{e_4\} = \varnothing$ |
| 3 | $i_n = Objects_U(i_c, (A_d, n_d))$ | $i_n = ((1,2), (2,2))$ |
| 3 | $A_f = Objects_F(i_n, d)$ | $A_f = \varnothing$ |
| 3 | $t_{tra} = Corr_{Tra}(t_f)$ | $t_{tra} = $ eighth row of Table 17 |

**Table 18.** *Cont.*

| Phase | Step | Result/Comment |
|---|---|---|
| 3 | $i_c = i_n$ | $i_c = ((1,2),(2,2))$ |
| 3 | $r = Corr_{\mathcal{R}}(t_{tra})$ | $r =$ reaction rule *r2* |
| 3 | $result = Phase4(s_c, m_c, r, s_0, m_0, n_x)$ | Phase 4 completed without error |
| 3 | $s_c, m_c, n_x = result$ | $s_c = $ <br> $m_c = \{(0,3),(3,0),(4,1),(1,4),(2,2)\}$ <br> $n_x = 5$ |
| 3 | End—ok | |
| 1 | $W_c, s, m_s, i_s = result$ | $W_c = \varnothing$ <br> $s = $ <br> $m_s = \{(0,3),(3,0),(4,1),(1,4),(2,2)\}$ <br> $i_s = ((1,2),(2,2))$ |
| 1 | $d = d + 1$ | $d = 3$ |
| 1 | $S_r = S_r \cup \{((d-1, s, m_s, i_s))\}$ | $S_r = S_r \cup \left\{\left(2, \quad, \{(0,3),(3,0),(4,1),(1,4),(2,2)\}, ((1,2),(2,2))\right)\right\}$ |
| 1 | End—ok | |

The result of the algorithm contains a state where both robots are in *AT2* area despite the fact that the walk has not passed through such state. A Gantt diagram for this scenario is shown in Figure 10. Both robots are performing actions *r1* and *r2* in parallel.

| | Time | | |
|---|---|---|---|
| | 1 | 2 | 3 |
| O1 | r1 | r2 | |
| O2 | r1 | r2 | |

**Figure 10.** A Gantt diagram for the scenario from the second example. Activities marked as *t* in the row preceded by *Ox* denote involvement of the element *x* (*x* is the unique identifier of a task element given at its first appearance or at the beginning of a scenario) during the activity *t*. Only elements that are active objects are included in the diagram.

Functions tupled with each state allow to "track" task elements between states. For example, the function $m_s = (0,0), (1,1), (2,2), (3,3), (4,4)$ for the state at time 0 indicates that the object tagged with the unique identifier 2 (the argument of $m_s$ function) is represented by the vertex with identifier 2 (the value of $m_s$ function for argument 2). The support of a bigraph itself does not track its elements between transitions, as can be seen by comparing the state of the system at time 0 and time 1. For example, knowing that there is one area of each type, we have no doubt that a vertex with the control *AT1* represents the same object in both states even though the support element assigned to each vertex is different between states. However, we do not have such certainty for vertices with controls

of the type *B*. Unique identifiers point to unique objects between states, even if those objects have changed the controls representing them.

Here is an example based on the elements of set $S_r$ from Table 18 how to use a unique identifier mapping. For the state at time 1, the UI with the value of 3 points to the vertex with identifier 1. This means that it is the same task element that in the state at time 0 is represented by the vertex with identifier value of 3 and the same element that at time 2 is represented by the vertex with support 0.

### 3.3. Example of Verifying the Fulfillment of Non-Functional Requirements

The last example is intended to demonstrate how non-functional requirements can be defined for systems designed using our methodology and determine whether these requirements have been satisfied.

For this example, we will define a task of relocating items in a warehouse. The goal of this task is for two robots to deploy items of different types from the warehouse to unloading areas. The initial state of the task is depicted in Figure 11. The interpretation of each control is shown in Table 19. Six reaction rules are defined for this system; all of them are listed and described in Table 20. For this example, the graphical representation of reaction rules is omitted because it will not be relevant.

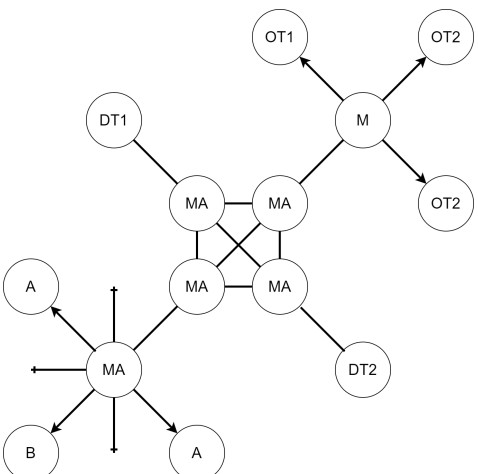

**Figure 11.** The initial state of a system in the example of checking whether non-functional requirements are met.

**Table 19.** Interpretation of controls in the example of checking whether non-functional requirements are satisfied.

| Control | Real World Object |
| --- | --- |
| *A* | Robot |
| *MA* | Warehouse area—robots can move between them. |
| *B* | Beacon—indicates the warehouse area where robots should return after relocating objects. |
| *M* | Warehouse—it stores objects to be moved. |
| *OT1* | Object of type 1 |
| *OT2* | Object of type 2 |
| *DT1* | Type 1 unloading area—the location where objects of type 1 are to be relocated. |
| *DT2* | Type 2 unloading area—the location where objects of type 2 are to be relocated. |

**Table 20.** System reaction rules for the example of checking whether non-functional requirements are satisfied. A value in the third column is the amount of time required to execute a rule.

| Label | Description | $\Delta T$ |
|---|---|---|
| mov | Moving a robot between warehouse areas. | 1 |
| stay | A robot remains in the warehouse area where it is located. | 1 |
| get1 | A robot retrieves a type 1 object from the warehouse. | 2 |
| get2 | A robot retrieves a type 2 object from the warehouse. | 2 |
| set1 | A robot deposits a type 1 object into an unloading area. | 2 |
| set2 | A robot deposits a type 2 object into an unloading area. | 2 |

The state space for the system consists of 666 states (vertices) and 5325 transitions (arcs). Due to the size of this example, the graphical representation of the state space and elements of the tracking transition system will not be presented. It is worth discussing here the increase in the size of a state space as the number of system elements increases. If one were to expand the current system to three robots, two type 1 objects, and three type 2 objects, the number of states increases to 5765 and the number of transitions to 70,701. Such a significant increase in the size of a system suggests that it is reasonable to consider ways of limited construction of a state space that will remain useful in later stages of the development of behavior policies.

Moving on to behavior policies for the agents in the task above. First walks solving the task are 15 steps long. However, these solutions are using only one robot, as can be observed in the action schedule presented in Figure 12. A mission performed using behavior policy based on such a walk takes 21 units of time.

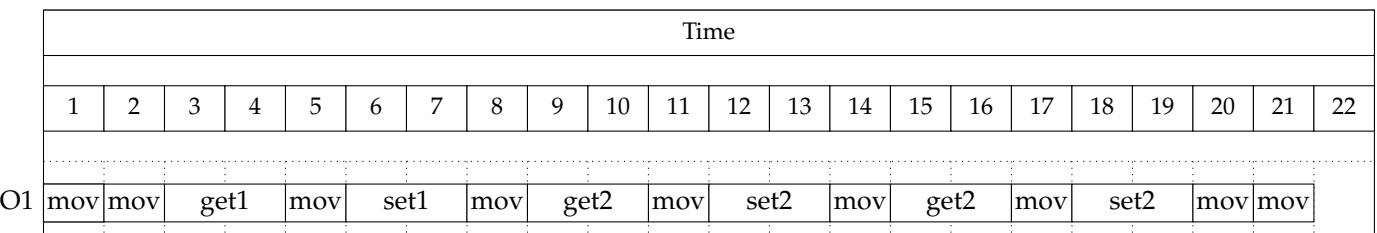

**Figure 12.** Schedule of actions for a scenario based on a walk of the length of 15 arcs.

### 3.3.1. Non-Functional Requirement—Length of a Mission

Now let us assume that one of the non-functional requirements imposed on the task is to limit the length of a mission to the maximum of 20 units of time. There is no walk of the length 15 that satisfies this requirement. Knowing that the current solutions use only one robot we can try to improve them by extending the walks to 18 steps. This way the second robot can move one of the objects to an unloading area. A schedule of actions constructed with a walk of 18 steps is presented in Figure 13.

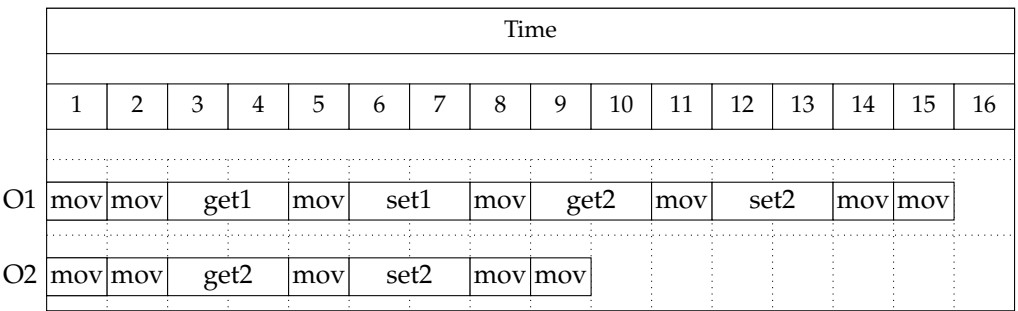

**Figure 13.** A schedule of actions for a scenario created with a walk of the length of 18 arcs.

It is important to note that simply lengthening a walk does not guarantee an improved result. For example, if the walk underlying the schedule in Figure 12 is lengthened by three transition functions, all corresponding to the reaction rule *stay*, it will not yield any improvement in the quality of a solution.

Checking whether non-functional requirements are fulfilled should be done in Phase 1 after Phase 3 has been successfully completed. This step is not shown on Scheme 4 but this is how it was implemented in [39].

### 3.3.2. Non-Functional Requirement—Collision Avoidance

Another example of a non-functional requirement will be related to safety of mission execution. This time we impose a requirement that there should be no collisions between robots that are in the process of moving objects.

One of the advantages of using bigraphs is that they allow one to define patterns to be found in other bigraphs. These patterns are of "minimal satisfying phenomenon" type. One cannot define an "all but" type pattern in bigraph notation. In other words, you can define a pattern like "minimum three people in a room" but you cannot define a (single) pattern that detects "less than three people in a room".

Let us assume that a collision-free mission will be guaranteed if the robots moving the objects are not in the same area. Such a requirement can be defined as "if two robots, at least one of which is moving an object, are in the same area then the scenario is unacceptable". Bigraph patterns able to detect such a situation are shown in Figure 14.

Identical to the previous non-functional requirement, this requirement can be verified in Phase 1 after a successful completion of the Phase 3.

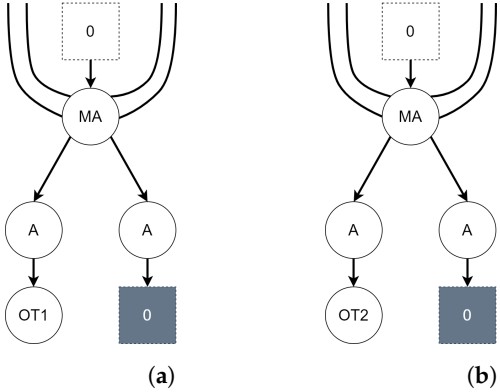

(**a**)                    (**b**)

**Figure 14.** Bigraph patterns to detect whether a collision between robots may occur during a scenario. The two patterns differ only in type of the relocated object. (**a**) The first pattern. (**b**) The second pattern.

### 3.4. Memory Complexity

As we have already mentioned, the size of a system grows much faster than the number of task elements. The same is true for the memory complexity of matrix multiplication operations described in Section 2.4. We have tested how limiting the number of results of convolution operation affects memory usage of the tool [40]. All measurements were done using multi-threaded F# implementation on a PC with 64 bit Ubuntu 20.04 operating system installed and the previous example regarding non-functional requirements was used for testing. We carried out three different tests reducing the number of results to 500, 10,000, and leaving the number of results unlimited. In the first case, the peak memory usage was about 700 MB before walks of the length of 15 arcs were found. The second case resulted in memory consumption around 15 GB before similar walks were found. The last case did not succeed on a machine with 64GB of RAM.

To deal with the memory complexity, we propose three methods to reduce the number of results:

- *First N*—a result of the convolution operation performed during a matrix multiplication is limited to the first $N$ results. This way of searching for behavior policies is suitable when the first results found satisfy non-functional requirements;
- *Best N*—a result of the convolution operation is constrained to the $N$ best results evaluated using an evaluation function (discussed below). This method of searching for walks is useful when a desired walk should have a certain length;
- *All*—the result of a convolution operation is not constrained in any way. Useful only for small systems to verify model correctness.

In the case of *best N* method, there is a need is to define an evaluation function for partial solutions. We propose a SAT configuration evaluation function based on the involvement of task objects. The evaluation function returns a higher score the more objects are involved equally. The formula for calculating the evaluation function value can be expressed as below:

$$E(i) = m = 0, \forall_{(o_{id}, t) \in i} m = m + \frac{t}{t_{max}} \qquad i \in I$$

$t_{max} --- $ The largest engagement of any object.

Table 21 shows the values of the proposed evaluation function for a few example SAT configurations.

**Table 21.** Partial solution evaluation function values for random SAT configurations.

| $i$ | $t_{max}$ | $E(i)$ |
|---|---|---|
| $((1,2),(2,2),(3,2))$ | 2 | 3 |
| $((1,6),(2,0),(3,0))$ | 6 | 1 |
| $((1,2),(2,4),(3,0))$ | 4 | 1.5 |
| $((1,1),(2,1),(3,4))$ | 4 | 1.5 |
| $((1,3),(2,2),(3,1))$ | 3 | 2 |
| $((1,1),(2,1),(3,0))$ | 1 | 2 |

The prepared tool [40] for walk construction offers six strategies for finding solutions:

- *All first found*—Returns all walks leading to the goal state with the shortest length;
- *First N found*—returns all walks leading to the target state. The matrix multiplication operation is constrained by *first N* method;
- *First N best found*—returns all walks leading to the target state. The matrix multiplication operation is constrained by *best N* method;
- *All up to a certain length*—returns all walks leading to the target state of a length no greater than a given value;
- *First N up to a certain length*—returns all walks leading to the target state with a length no greater than a given value. The matrix multiplication operation to find walks in a state space is constrained by *first N* method which results in each set of walks of the same length being allowed to have a count of at most $N$ elements;
- *Best N up to a certain length*—returns all walks leading to the target state with a length no greater than a given value. The matrix multiplication operation to find walks in a state space is constrained by *best N* method which results in each set of walks of the same length being allowed to have a count of at most $N$ elements.

We summarized all of the above strategies in Table 22.

**Table 22.** Summary of the proposed strategies for finding walks in a state space. The second column denoted as MNoR stands for Maximum Number of Results. The value of $N$ is equal to a number of results of the same length. The value of $L$ is equal to the maximum length of a result.

| Strategy | MNoR | Pros | Cons |
|---|---|---|---|
| All first found | Unlimited | Perfect for assuring correctness of the model as this strategy gives all existing walks to the desired destination state. | Unfeasible for anything but small systems due to large memory consumption. |
| First N found | N | The fastest of all strategies since it does not sort results and can shrink an output of convolution operation. Perfect when the quality of a result is not important or when all results are expected to have similar quality. | Does not care about quality of returned results at all. |
| First N best found | N | With a good evaluation function this strategy can return the best results. Perfect when model has already been validated and the developer is looking for a behavior policy of a certain quality. | Slower than *first N found* since results are sorted with an evaluation function. |
| All up to a certain length | Unlimited | Gives a glimpse of how the length of a walk impacts the way a mission is executed. Since it is an extension of *all first found* it allows for throughout correctness testing. | Only for tiny systems. This is the most memory consuming strategy because it not only returns all found results but the search is continued until results have specified length. |
| First N up to a certain length | $N \times L$ | Allow for insight into how the length of a result impacts the way a mission is executed. Very fast as it is an extension of *first N found*. | Does not care about the quality of returned results at all. |
| Best N up to a certain length | $N \times L$ | It gives good insight how the quality of results varies with the length of a walk. Perfect when the developer is looking for a behavior policy that he or she has no expectations about. | It is slower than *first N found up to a certain length* strategy due to sorting of results. |

## 4. Discussion

In this paper, we presented an algorithm to verify multi-agent system models based on tracking bigraphical reactive systems. Our algorithm can detect incorrectness of a model and unfulfillment of non-functional requirements. The algorithm considers a model to be incorrect if activities planned to be executed in parallel are not independent of each other. In this article, we presented two examples of utilizing the algorithm to check if a behavior policy meets non-functional requirements regarding time and safety of task execution. We also demonstrated how to generate successive states of a scenario, which is a task realization using a selected behavior policy, based on the the behavior policy. Finally, we discussed memory complexity of operations essential to behavior policies generation and proposed a few ways to reduce it. One of the suggested methods is to limit results to a certain number of the best ones. We gave an example of an evaluation function that allows ranking partial results (in our case, these are behavior policies that when executed do not meet functional requirements). The evaluation function is applicable to tasks of any kind and size.

This work complements our previous publication, which focused solely on designing multi-agent systems with tracking bigraphs. The methodology enables the design of a broad range of systems from warehouse robots to drone swarms performing a task without human intervention. One can also consider designing software systems where programs act as agents and operations performed by these programs could represent transition functions. The functional programming paradigm intuitively fits this kind of design.

The main drawback of our methodology is the lack of adaptability of behavior policies. This means there can be no deviation from scheduled actions when executing a behavior policy. It also means that agents in a modeled system have to be fully controllable in the real world. The biggest drawback of the algorithm presented in this article is that it verifies the correctness of a model looking for errors in a single behavior policy. Thus, the more behavior policies that are checked, the more confident we are that the model is correct.

As for directions of further development, the primary goal should be to improve the generation speed of tracking reactive systems as it is the main limitation of the methodology right now. One way to achieve it is to develop a method of partial construction of a tracking bigraphical reactive system that consists of bigraphs necessary to manufacture a good quality walk in state space. If the method of reducing the number of states is automatic, i.e., it will not require the designer to specify bigraphical patterns, it is going to significantly speed up the development of behavior policies. Right now our method can only be applied to relatively small systems because the explosion of states makes it impossible to efficiently search for walks in the state space of a modeled system.

**Author Contributions:** Conceptualization, P.C. and Z.Z.; methodology, P.C. and Z.Z.; software, P.C.; validation, P.C.; formal analysis, P.C.; investigation, P.C.; resources, P.C.; data curation, P.C.; writing—original draft preparation, P.C.; writing—review and editing, Z.Z.; visualization, P.C.; supervision, Z.Z.; project administration, Z.Z. Both authors have read and agreed to the published version of the manuscript.

**Funding:** This research received no external funding.

**Institutional Review Board Statement:** Not applicable.

**Informed Consent Statement:** Not applicable.

**Data Availability Statement:** Data sharing not applicable.

**Conflicts of Interest:** The authors declare no conflict of interest.

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
