# Peer review of "Design and Verification of Multi-Agent Systems with the Use of Bigraphs"

_applsci, doi:10.3390/app11188291_

Round 1

Reviewer 1 Report

This paper presents an approach, using tracking bigraphs, to model multi-agent systems and (importantly) how they can evolve over time. It is very closely related to the paper cited as [23] (Tracking bigraphs applied to UAV scenarios).

While I do think there are interesting ideas in this paper (in particular tracking bigraphs are not very common in the literature, so it's interesting to see what they can do), I found the presentation to be very challenging to read which really limits the usefulness of this paper (in it's current form).

I recommend significant rewrites to the introduction and background sections (more details below), to bring out the core ideas and help guide the reader. Because I struggled with the background, I have not been able to fully dive into the algorithm details.

Detailed Comments
------------------

- I found the overall aims/goals (model correctness, simulation, and non-functional reqs) of the paper to be interesting, but really struggled to unpick exactly where they occur in the paper and what the outcomes are. For example, "simulation" doesn't seem to be mentioned again after l101 in the intro. Likewise you say "security" as a non-functional requirement, and again it doesn't seem to be mentioned elsewhere in the paper.

- It was great to see many open-source software citations and that you are making the tooling available---this is a big plus! A brief look at the tooling suggests it builds on BigraphER so suggest a citation to that tool (there's a CAV paper) would be appropriate.

- While there's a lot of background on multi-agent systems, there is less on Bigraphs and their uses. I suggest expanding on this in the intro. In particular there are papers on Multi-agent systems in bigraphs that are missing, e.g. Mansutti, Alessio, Marino Miculan and Marco Peressotti. “Multi-agent Systems Design and Prototyping with Bigraphical Reactive Systems.” DAIS (2014).

- A short discussion on why tracking bigraphs over standard bigraphs would also aid the paper as these are much less known.

- I think throughout section 2, interleaving it with an example (such as the one later in the paper) would *really* help! There's a lot of definitions and it's really hard to get an intuitive feel for how this works in real terms, esp behaviour policies etc.

2.2.
 - Possibly make it clear you work with concrete bigraphs (as is required for tracking I guess?).
 - O<n,Y> is missing an =
 - You never seem to use instantiation maps (eta) in the examples, so you can probably simplify your reaction rule defn (by assuming it's id)
 - Reaction rule is m -> I, but I = <m, X>; suggest you use another ordinal (else it looks like num sites = num regions). Be explicit these are /bigraphs/ ("redex" doesn't say the /type/ of object, i.e. on l163).
 - Suggest link id's on the figure for clarify

 - Can you clarify how to map from BRS to TTS, i.e. is Agt = B? Red = R? Are labels the reaction names or something else (It's not clear).

 - A notion of time appears in 2.3, but it's not clear how this relates to the reaction rules (they don't currently specify a time that they take etc). Suggest a whole sub-section on how time is modelled would really help.Possibly make it clear you work with concrete bigraphs (as is required for tracking I guess?).

 - l214: You say you don't define + for the rest of the elements, but then you use it? Do you mean you don't show it, but that it is defined (elsewhere?)

   - l217: T seems badly typed: returns C, but Nat (i.e. 0) for f_null. Should it be C X N -> N? not (C X N -> C)

 - l297 mentions bigraph support, but this wasn't well defined (although is central to tracking so probably should be!)

 - I think there's value in giving the phases of your algorithm concrete names so it's easier to tell what they do. I.e. Phase 4 seems to be an independence check so perhaps call it independence checking (or conflict checking) etc. "phase 3" doesn't tell the reader much about what's happening, and makes the other algorithms difficult to follow.

 - I wonder if "algorithm" blocks, instead of flowcharts and I/O tables would be more appropriate here? The flowcharts take a lot of space, and the I/O tables could be included as text. (Note: this is more of a suggestion and a blocker to publication etc)

 - l482: Might be worth labelling stats on the table to show iso states clearly.

 - l482: Wasn't quite sure why it "must occur", can more explanation be added.

 - 3.4
   + It wasn't clear if the max numbers given, i.e. "memory consumption will be 15GB"  were empirically measured, or a theoretical maximum.
   + I really liked heuristic style searches and the fact you have tool support, but this section seems unfinished: you have 6 strategies but no analysis of their trade-offs etc (which would really improve the paper!), e.g.:
     - What memory it saves,
       + How it affects results
       + Which methods have the best savings/results tradeoffs

Reviewer 2 Report

Overall the paper seems to be interesting and sound, however, in my opinion, it is still affected by some problems.

First of all, in some parts, the clarity and editorial quality of the paper weaken. As a consequence, such parts result to be quite difficult to read. Therefore, I would suggest to carefully improve the prose of writing in order to make this paper easier to read.

Furthermore, presentation aside, by reading the paper, it still was not entirely clear what to expect with the direction of the article. Indeed, the contribution proposed in this paper has been only marginally compared and contextualised with respect to the state of the art. As a result, it is extremely difficult to understand the novelty/contributions introduced by the paper. The aforementioned aspects should be carefully addressed before the paper can be considered any further.

The paper should be better compared and contextualized with respect to the state of the art. I want suggest this paper to authors:

- Flora Amato, Francesco Moscato, Vincenzo Moscato, Francesco Pascale, Antonio Picariello, An agent-based approach for recommending cultural tours, Pattern Recognition Letters, Volume 131, 2020, Pages 341-347, ISSN 0167-8655

The paper results to be quite difficult to follow, due to the massive (and sometimes not adequately described) use of mathematical notation. The use of mathematical notation should always be supported by an appropriate informal description. By doing this, the paper may be easier to read and follow. 

Finally, a thorough proofreading would be suggested, since in the paper there are some typos and formatting issues.

As remarks:

  • The paper should be better compared and contextualized with respect to the state of the art.
  • In some parts of the paper, the clarity and editorial quality of the paper weaken. As a consequence, such parts result to be quite difficult to read. Therefore, I would suggest to carefully improve the prose of writing in order to make this paper easier to read.
  • Each figure should be properly defined within the text and must be improved in quality.
  • Equations should be numbered appropriately and referenced within the text.
  • An accurate proofreading is strongly recommended.

Round 2

Reviewer 1 Report

Thank you for improving the quality of the paper, the changes to the introduction and the background have made it significantly easier to make sense of the details.

While the mathematics is still very dense, the worked examples make it possible to figure out what is going on, and I don't see any particular issues with the algorithms.

I believe this paper can be accepted with a few very minor edits. Importantly for section 2.5 it would be good to ensure the diagrams (flowcharts) for the algorithm phases are kept close to the description (sometimes you had to move a few pages to see the flowchart); this is only an editing issue rather than content.

Some minor changes for the authors:

- Wasn't clear where the Del (upside down delta) operator comes into l272, I thought these would be reaction names? If not a description of the labels and why they look like this would be useful.
- Table 1 (near end) r = "regula reakcji r2"? Not sure what this is meant to be?
- Figure 10, what is Czas? (Needs translated? "Time" I guess?)

Reviewer 2 Report

all suggestions have been applied

Author Response

Dear Reviewer

Thank you once again for your comments and suggestions that have undoubtedly helped us improve the quality of the article.